# [Re] FairDICE: A Fair Tradeoff in Multi-objective Offline RL

**Peter Adema, Karim Galliamov, Aleksey Evstratovskiy, Ross Geurts**
*{peter.adema, karim.galliamov, aleksey.evstratovskiy, ross.geurts}@student.uva.nl*
*Informatics Institute*
*University of Amsterdam*

**Reviewed on OpenReview:** *https://openreview.net/forum?id=Tr6MBtOhAj*

## Abstract

Offline Reinforcement Learning (RL) is an emerging field of RL in which policies are learned solely from demonstrations. Within offline RL, some environments involve balancing multiple objectives, but existing multi-objective offline RL algorithms do not provide an efficient way to find a fair compromise. FairDICE seeks to fill this gap by adapting OptiDICE (an offline RL algorithm) to automatically learn weights for multiple objectives to e.g. incentivise fairness among objectives. As this would be a valuable contribution, this replication study examines the replicability of claims made regarding FairDICE. We find that many theoretical claims are supported, but an error in the code reduces FairDICE to standard behaviour cloning in continuous environments, and many important hyperparameters were underspecified. After rectifying this, we show in experiments extending the original paper that FairDICE can scale to complex environments and high-dimensional rewards, though it can be reliant on (online) hyperparameter tuning. We conclude that FairDICE is a theoretically interesting method, but the experimental justification requires significant revision.

## 1 Introduction

Many tasks are best defined not in terms of input-output pairs but in terms of rewards or goals to be achieved, and Reinforcement Learning (RL) is often a natural choice for modelling them (Sutton & Barto, 2018). In many real-world domains, such as medicine or robotics, however, the cost of training a policy from scratch in a live ('online') environment is often unacceptable, necessitating the use of offline RL. Additionally, real-world applications of RL often involve complex environments with multiple, occasionally conflicting goals; since standard RL requires a single reward as its target, choices must be made about how to combine diverse goals into one. A straightforward method would be to take a weighted sum of the rewards, but determining weights such that the resulting policy behaves 'fairly' and does not focus on one reward at the expense of others is non-trivial (Hayes et al., 2022). Ensuring such fairness can be essential when the objectives represent medical outcomes, or the interests of different population groups (Smith et al., 2023), and many environments where fairness is key are also those where online evaluation is challenging or carries unacceptable risk.

Kim et al. (2025a) derive the FairDICE algorithm for offline multi-objective RL as a variation of OptiDICE (Lee et al., 2021), proposing to indirectly optimise a concave, non-linear objective which would ensure the returns of each objective are of similar magnitude (accomplished via a learned linearisation of the objectives, see Sec. 2.2). As such, FairDICE could provide a baseline for fairly balancing multiple objectives in offline RL. Kim et al. (2025a) claim that FairDICE is an effective method and purport to show this using both discrete and continuous environments.

This replication study aims to investigate to what extent these claims can be verified and to clarify details of the public implementation of FairDICE not described in Kim et al. (2025a). We examine claims in both discrete environments (where theoretical properties such as the effect of $\alpha$ and $\beta$ are verified on simple tasks) and continuous environments (where FairDICE is compared against preference-conditioned baselines on the D4MORL benchmark (Zhu et al., 2023)).

Beyond direct replication, we extend the evaluation to four settings not covered by the original paper: environments with high-dimensional rewards (100 objectives), image-based observations, datasets biased to a specific objective, and learning from unnormalised, negative rewards and returns.

In this replication, we uncover a significant discrepancy between the FairDICE algorithm as described and as implemented. A broadcasting error in the public code causes the policy loss in continuous environments to ignore the learned importance weights entirely, reducing FairDICE to standard behaviour cloning (BC). As this error still produced seemingly valid policies, it was not detected in the original (Kim et al., 2025a). Consequently, because the critic had no influence on the trained policy, the effects of hyperparameters were masked, and FairDICE appeared highly (unusually) robust across different hyperparameter settings (in continuous environments). We confirm this finding through statistical testing and correspondence with the original authors, who acknowledged the issue was present in published (and unpublished) experimental code.

After correcting the implementation, we show that the theoretical properties of FairDICE are largely supported by experiments, and that the algorithm demonstrates an ability to learn fair policies that standard behaviour cloning cannot. However, the corrected algorithm is highly sensitive to the regularisation strength $\beta$, with no clear pattern for how to select $\beta$ across environments. Such unpredictable behaviour undermines the idea that FairDICE can be applied without hyperparameter tuning – which is challenging in an offline setting (though not impossible, see e.g. Paine et al., 2020). We further extend the original evaluation to environments with high-dimensional rewards (100 objectives) and image-based observations, where FairDICE shows promising scalability, as well as to biased datasets, where its robustness has limits. Taken together, our findings suggest that FairDICE as a contribution has well-supported theoretical claims, but its experiments would require substantial revision to accurately reflect the true strengths and weaknesses of the method.

In this report, Sec. 2 provides an overview of some relevant background material, with Sec. 2.3 listing the claims from Kim et al. (2025a) to be replicated. Afterwards, Sec. 3 summarises the experimental method, with two points of interest in the public FairDICE implementation being described in Sec. 3.3. Finally, Sec. 4 and Sec. 5 provide experimental results and a discussion thereof.

## 2 Background

### 2.1 Offline Reinforcement Learning

Reinforcement Learning (RL) is often defined on a Markov Decision Process (MDP, Puterman, 1990) of the form $\mathcal{M} = \langle \mathcal{S}, \mathcal{A}, \mathcal{R}, T, p_0 \rangle$, where $\mathcal{S}$ is the set of all (discrete) states, $\mathcal{A}$ is the set of all possible actions, $\mathcal{R}$ is a (stochastic) reward function from state-action pairs to real-valued rewards, $T$ is the (stochastic) transition function from state-action pairs to next states and $p_0$ is the initial state distribution over $\mathcal{S}$. Within this MDP, RL introduces a policy $\pi$ which provides a distribution over actions for each state (Sutton & Barto, 2018). A common RL task is *optimal control*, whereby we seek to find a policy $\pi^*$ which maximises the expected discounted return $J(\pi)$ received from $\mathcal{M}$. If we define $R_t$ as the reward received at timestep $t$ of the MDP and $\gamma \in [0,1)$ as the discount factor, then we can define this target as $J(\pi) = \mathbb{E}_\pi \left[ \sum_{t=0}^\infty \gamma^t R_t \right]$.

Classical RL algorithms work *online*, in that training and evaluation is done with access to the underlying environment / MDP (Sutton & Barto, 2018). However, for some domains (e.g. medical or logistical) it is impractical to let a process train in a real-world environment, and policies are instead trained using a dataset $\mathcal{D} = \{(s_t, a_t, s_{t+1}, R_t)\}$ of state, action, next state and reward tuples gathered using existing policies. RL algorithms which can function using only $\mathcal{D}$ (without access to $\mathcal{M}$) are referred to as *offline* RL algorithms. The simplest offline RL algorithm is Behaviour Cloning (BC), whereby the policy $\pi$ is trained to mimic the actions taken in the dataset $\mathcal{D}$ (Kumar et al., 2022). More complex algorithms aim to estimate some of the mechanics of $\mathcal{M}$, so as to be able to express a preference for which actions should be imitated or avoided by the new policy. Many optimal control algorithms for offline RL exist (e.g. Kumar et al., 2020; Kostrikov et al., 2021), but relevant for this reproduction is primarily the OptiDICE algorithm proposed by Lee et al. (2021). The basic OptiDICE algorithm (on which FairDICE is based) learns a critic $\nu$ from $\mathcal{D}$ which estimates the value of each state, and uses this critic to calculate weights $w^*(s, a)$ expressing whether a certain state-action pair should be imitated more or less by an optimal policy (for details, see Nachum et al., 2019 and Lee et al., 2021). Policies are then learned via weighted behaviour cloning: $\pi = \arg\max_{\pi'} \mathbb{E}_{(s,a)\sim\mathcal{D}} [w^*(s, a) \log \pi'(s|a)]$, where the strength of $w^*$ is controlled by a hyperparameter $\beta$, with a large $\beta$ constraining $w^*$ to be near 1.

## 2.2 Multi-objective RL and FairDICE

OptiDICE, like most traditional RL algorithms, is formulated in terms of maximising a scalar return $J(\pi)$, but realistic environments often involve balancing $J_i(\pi)$ for multiple, sometimes conflicting objectives $i$ (Hayes et al., 2022). Adapting such multi-objective environments for use with traditional RL algorithms often involves *scalarisation*, whereby the multiple rewards are combined into a single scalar utility. However, the choice of scalarisation directly influences the optimisation target and thereby the types of policies which are optimal. Furthermore, the desired behaviour of an optimal policy can often best be described by a non-linear combination of rewards (e.g. certain objectives must reach a minimum value, or other objectives have limited utility above a certain value), but such a non-linearity would contradict the assumptions of traditional RL methods relying on linearly decomposing the return $J$ into per-step (TD) predictions (Roijers et al., 2018). A common use-case of non-linear utilities lies in assessing 'fairness' of policies in multi-objective environments; whether a policy balances several objectives or focuses on a subset at the expense of the rest (Ramezani & Endriss, 2009; Fan et al., 2022). When RL algorithms are applied to make real-world decisions, unfairness in balancing objectives can lead to biased or undesirable outcomes (Smith et al., 2023).

To assess the fairness of a set of returns $J_i(\pi)$, the Nash Social Welfare (NSW) function from economic theory (Kaneko & Nakamura, 1979), defined in this context as $\mathrm{NSW}(\pi) = \sum_i \log J_i(\pi)$, is often used as a simple metric. Due to its non-linear nature, NSW cannot be used directly as a scalarisation method for traditional algorithms. A possible approach to resolve this would be to use a linear weighting vector $\mu$ and to define the surrogate target $\hat{J}(\pi) = \sum_i \mu_i J_i(\pi)$ on which a traditional RL algorithm can be trained, where for an appropriate choice of $\mu$ the optimal policy for $\hat{J}$ might also have a high NSW (Zhu et al., 2023). With access to the environment for evaluation, it is then possible, though computationally expensive, to examine policies trained using a variety of options for $\mu$ and select one with a high NSW. However, this process (evaluating policies) is not possible in the truly offline setting for which we seek to devise algorithms, and searching through the space of possible values for $\mu$ is also impractical with high-dimensional rewards.

To improve upon this idea, Kim et al. (2025a) derive FairDICE as an adaptation of OptiDICE, which is able to *learn* the optimal $\mu$ such that maximising $\hat{J}$ also optimises for a sum of non-linearities $u_i$: $\sum_i u_i(J_i(\pi))$, all without online evaluation. FairDICE proposes to learn the preference vector $\mu$ in addition to a critic $\nu(s)$ (e.g. a multi-layer perceptron), where an additional regularisation term on $\mu$ is added to the OptiDICE loss to incentivise rewards to be distributed according to the selected non-linearities $u_i$ (e.g. evenly balanced, for NSW). Kim et al. (2025a) use $\alpha$-fairness as their non-linearity: $u_i(x) = \frac{1}{1-\alpha}x^{1-\alpha}$ where $\alpha \neq 1$ and $u_i(x) = \log(x)$ for $\alpha = 1$. The sum $\sum_i u_i(J_i(\pi))$ then reduces to NSW for $\alpha = 1$ (used for most experiments), a summation (as in utilitarian welfare) at $\alpha = 0$ and a minimum (as in max-min welfare) as $\alpha \to \infty$. Outside of this regularisation term and linearisation scheme, FairDICE is identical to OptiDICE and uses weighted behaviour cloning for policy learning. For more details, refer to Kim et al. (2025a) and see Appendix A.

## 2.3 Claims and extensions

Using this framework, Kim et al. (2025a) run a series of experiments using continuous and discrete environments. The experiments using discrete environments use toy tasks to verify theoretical properties of the FairDICE algorithm in practice, while the experiments using continuous environments aim to scale to more realistic MuJoCo environments and compare with prior work by Zhu et al. (2023) done on the D4MORL offline RL benchmark. These experiments are what this reproducibility study will focus on verifying. We identify the following claims in and propose the following extensions of Kim et al. (2025a):

1. Using **discrete** environments, the following theoretical aspects of the algorithm are claimed:

    Claim 1.1 FairDICE learns a balanced policy using trajectories from a uniformly random policy, while reaching each objective more frequently than a random policy.

    Claim 1.2 Changing $\alpha$ interpolates the trained policy between utilitarianism and min-max fairness, while changing $\beta$ interpolates between welfare maximisation and behaviour cloning.

2. Using **continuous** environments, the following claims are made of practical performance:

    Claim 2.1 FairDICE performs consistently across a range of values for $\beta \in [0.0001, 1]$.

    Claim 2.2 FairDICE performs competitively with baselines on the D4MORL benchmark, lying on the Pareto front of preference-conditioned policies as defined by Zhu et al. (2023).

3. Extending the original paper, we will also examine the following four claims about FairDICE, which should follow from the above claims or from public OpenReview comments (Kim et al., 2025b):

  Ext. 3.1 FairDICE remains effective on negative returns using a piecewise alternative to log as $u_i$.

  Ext. 3.2 FairDICE can learn policies competitive with the baselines using only training data from policies biased to specific rewards (e.g. a policy which primarily achieves one objective).

  Ext. 3.3 FairDICE can improve over the data policy in an environment with many (100) rewards.

  Ext. 3.4 FairDICE can scale to a complex image-based environment such as Minecart-RGB.

Regarding the theory behind and derivation of FairDICE, we believe the derivation to be self-consistent (save for a likely typo; see Appendix L) and in line with the related material cited by Kim et al. (2025a). However, we do not focus on examining the mathematical validity of the (Fair)DICE algorithm. Instead, we hope to provide insight into the practical utility of FairDICE for (fair) offline reinforcement learning by examining these claims and extensions.

## 3 Method

Accompanying FairDICE is a GitHub repository (Kim et al., 2025c) which contains an implementation of the core FairDICE algorithm, as well as code for training FairDICE in the continuous environments using the D4MORL dataset. However, code for running the baselines in the continuous case, as well as all code related to the discrete environments, was not included in the published code and had to be recreated (though, through a later correspondence with the authors, we did obtain code for the discrete environments). Furthermore, during the review of the existing FairDICE code, two inconsistencies were discovered with the algorithm as described by Kim et al. (2025a), namely a broadcasting mistake when calculating the policy loss and an additional gradient penalty term on $\nu$ in the critic loss. We therefore run all experiments using both the FairDICE implementation as provided in code and as described in the paper.

### 3.1 Environments

For the experiments using discrete environments, two toy examples were described in Kim et al. (2025a). First is **MO-Four-Rooms**, a simple environment with three goals. The agent starts in the top-left room and must move to one of the goals in the other three rooms, receiving a one-hot reward per goal. Secondly, **Random MOMDP**, a randomly generated multi-objective MDP, also with three one-hot-rewarding goals.

The primary results, detailing more realistic performance measures, were obtained using the multi-objective variant of the D4MORL offline RL benchmark from Zhu et al. (2023). The benchmark consists of six MuJoCo-simulated control environments, where the agent must move forward while accomplishing a secondary goal (two rewards). The environments are named **Hopper**, **Swimmer**, **Walker2d**, **Ant** and **HalfCheetah**, where Hopper also has a three-objective variant with an additional reward for vertical height.

Finally, we add two environments to evaluate the generalisability of FairDICE. First, **MO-GroupFair**, a (novel) miniature model of societal unfairness with 100 individuals (each representing one reward) who are members of 5 groups. The agent (e.g. a government) is given 1 unit of reward and 7 random options for distribution amongst the groups, but groups which were rewarded more in previous iterations gain favourable future options, naturally leading to escalating unfairness. Secondly, we include **MO-Minecart-RGB**, a three-reward environment from Abels et al. (2019) in which an agent must collect two types of ores while maintaining energy efficiency. Observations are RGB images, while the action space is discrete.

For all environments, a more detailed description can be found in the relevant section of Appendix B.

### 3.2 Datasets

For MO-FourRooms, two datasets generation strategies were used: one using a uniformly random policy as described in Kim et al. (2025a), and the other one using a biased MDP with goal visitations distributed at 80/10/10%. For Random MOMDP, a single dataset was collected using a policy with optimality level 0.5, as per original paper. Training on the D4MORL benchmark tasks was done using the existing dataset from Zhu et al. (2023), identically to Kim et al. (2025a). For MO-GroupFair, two datasets were collected using a uniformly random policy and a biased (unfair) policy (see Appendix B.4). Minecart-RGB, as a more challenging environment, had a dataset collected using an expert policy trained using PPO from Raffin et al. (2021), as described in Appendix B.5.

### 3.3 Discrepancies in public FairDICE code

#### 3.3.1 Incorrect policy loss for continuous environments

In the public FairDICE repository (Kim et al., 2025c), on line 135 of the main algorithm in FairDICE.py, the term $w^*(s, a) \log \pi'(a \mid s)$ of the policy loss is calculated using `stable_w * log_probs`. However, `stable_w` is a tensor of shape `(batch, 1)`, while `log_probs` is a tensor of shape[1] `(batch,)`: when these are multiplied, then by standard broadcasting rules (Ascher et al., 2001) the result is a tensor of shape `(batch, batch)`. In a later step, the result is summed over all axes in an attempt at a weighted sum, but due to the expanded shape, this actually results in all terms having the same weight, similarly to `stable_w.sum() * log_probs.sum()`. In other words, the code uses an outer product instead of a Hadamard product, and since `stable_w` was normalised in a previous step to have a mean of 1, a `stable_w.sum()` term would[2] be a constant (independent of $\nu$, $e(s, a)$ and hyperparameters such as $\beta$) equal to the current batch size. Since all actions are then weighted equally, the resulting policy loss is effectively equivalent to standard Behaviour Cloning (BC).

As the `log_probs` term for discrete environments would be correctly shaped `(batch, 1)`, we can therefore conclude that this BC-style loss was used for experiments in continuous but not discrete environments. Upon contacting the authors of Kim et al. (2025a) regarding this issue with the policy loss, the original authors confirmed it to be present in the published code and in the other, unpublished code used for continuous environments. The authors also stated in February that a new revision of the code and paper is underway to address this issue with the experiments in continuous environments.

For this report, many experiments in continuous environments were run with both FairDICE as implemented in Kim et al. (2025c) (suspected to be identical to BC) and a fixed version which correctly performs weighted behaviour cloning. All experiments in discrete environments were run using the correct policy loss.

#### 3.3.2 Additional gradient penalty on critic

Aside from the implementation of the policy loss, another unexpected part of the FairDICE algorithm code lies in an additional term (FairDICE.py:103-112) added to the critic loss which appears to incentivise the smoothness of $\nu$. When evaluating $\mathcal{L}(\nu, \mu)$, samples $(s_i, a_i, s'_i)$ are taken from the dataset to evaluate the TD-style error $e$, while starting states $\dot{s}_i$ are sampled[3] from $p_0$. A (singular) $\epsilon \sim \text{Uniform}[0, 1]$ is then sampled and used to perform linear interpolation between the initial and next states to obtain $\bar{s}_i = \epsilon \dot{s}_i + (1 - \epsilon) s'_i$. Then, with $\mathbf{p} \in \mathbb{R}^{\texttt{batch\_size}}$,

$$\texttt{grad\_penalty} = \lambda \, ||\mathbf{p}||_2^2, \quad p_i = \max\{0, ||\nabla_{\bar{s}_i} \nu(\bar{s}_i)||_2 - 5\}, \tag{1}$$

with $\lambda$ as a hyperparameter adjusting the strength of the gradient penalty. Notably, the default configuration of FairDICE in the public code has this gradient penalty enabled with $\lambda = 0.0001$, and provides no further explanation or motivation of this formula. Correspondence with the authors of Kim et al. (2025a) clarified this loss as incentivising the L2-smoothness of the critic as described in Kim et al. (2021) (different author) and Gulrajani et al. (2017), with the lack of mention being due to the original authors not noticing a meaningful impact of the gradient penalty. This apparent lack of impact can be explained by the policy loss in continuous environments incorrectly being simple behaviour cloning, meaning that the critic (and any adjustments to it) had no impact on the trained policy. Notably, the original authors also stated that this gradient penalty was only present for continuous and not discrete environments[4].

For this report, experiments in Sec. 4 using the fixed version of the policy loss for continuous environments examine values of $\lambda \in \{0, 0.0001, 0.1\}$ to investigate the impact of this smoothness term on performance.

---

[1]A likely explanation for this mistake is that Kim et al. (2025a) did not account for event shapes when porting discrete code to continuous. In `tensorflow_probability`, a discrete `Categorical` distribution has unit event shape `()`, meaning that evaluating the log probability of a discrete actions tensor shaped `(B, 1)` would produce `log_probs` of the same shape. However, a multivariate distribution over `A` actions has event shape `(A,)`, meaning that evaluating the log probability of an actions tensor shaped `(B, A)` would consume the trailing dimension and produce `log_probs` shaped `(B,)`, silently leading to the above mistake.

[2]This is slightly complicated by an additional `mask` for terminal states, but since `mask` is also shaped `(256, 1)`, it does not actually mask out actions from `log_probs` but instead causes `(mask * log_probs).sum()` to be slightly smaller than `batch_size`.

[3]In practice, Kim et al. (2025c) use the starting state from the trajectory which $s$ is a part of, not an independent sample.

[4]Which is logical, as interpolating a one-hot state representation would be very unintuitive.

### 3.4 Experimental setup and code

All experiments, except those on Minecart-RGB, MO-FourRooms and MO-MDP, parameterise the critic $\nu$ and the policy $\pi$ using an MLP with 1-4 layers and 512-768 hidden units, and train these MLPs using the Adam optimiser. Minecart-RGB uses a CNN described in B.5 and MO-FourRooms and MO-MDP both use tabular policy representation. Unless otherwise mentioned, all non-linearities $u_i$ are log (i.e. $\alpha = 1$). Precise model architectures and hyperparameters are provided in Appendix B. Both rewards and (continuous) states are normalised as in Kim et al. (2025a), with rewards being min-max normalised to [0, 1] and states being mean-std normalised (see Appendix C for details). For the D4MORL tasks, we compare FairDICE to three preference-conditioned offline MORL baselines from Zhu et al. (2023). **BC(P)** performs behavioral cloning conditioned on the state and preference vector $(s, \mu)$, learning a policy $\pi(a \mid s, \mu)$. **MODT(P)** extends Decision Transformer by modelling trajectories as sequences of (return-to-go, $s, a$) tokens concatenated with $\mu$. **MORvS(P)** is a feed-forward alternative that predicts actions from the current state, preference vector, and average multi-objective return-to-go. Following Kim et al. (2025a), we train and evaluate these baselines over a grid of preference weights and report performance (including NSW) as a function of $\mu$. Exact parameters for the three methods are given in Appendix E. For all configurations, models are trained for 5, 10, or 1000 seeds and subsequently evaluated using between 10 and 100 evaluation rollouts (depending on available compute). The primary metrics models are compared on are average NSW, calculated as $\frac{1}{N} \sum_{n=1}^{N} \sum_i \log J_i(\pi)$, Jain's fairness $(\sum_{i=1}^{n} R_i)^2 / (n \sum_{i=1}^{n} R_i^2)$ for rewards $i$ and $N$ evaluations. For consistency with Kim et al. (2025a), results on D4MORL are reported as the average *normalised* NSW, where the rewards which make up each return first undergo min-max normalisation as described in Appendix C. Code for the original FairDICE paper is available at ku-dmlab/FairDICE, while code for the experiments and graphs in this replication study is available at p-adema/re-fairdice.

## 4 Results

During replication, we encountered a major issue, namely the incorrect algorithm used to generate the original continuous results. Moreover, for experiments using the discrete MO-FourRooms and MO-MDP environments, Kim et al. (2025a) left important hyperparameters unspecified (and did not publish code in the official repository), making the results difficult to replicate. After correspondence with the original authors, however, code for these experiments was provided, and the results could be exactly replicated.

In continuous environments, we show that the public FairDICE implementation is effectively equivalent to BC, and that these BC-like performance figures are indeed the results shown in Kim et al. (2025a). When correctly implemented, the FairDICE algorithm is highly dependent on the correct tuning of $\beta$ (as well as the new hyperparameter $\lambda$, to a limited extent). In some environments (e.g. HalfCheetah), 'fixed' FairDICE outperforms baselines, but in others (e.g. Hopper) it performs identically or worse even when tuned[5].

Finally, we examine some extensions and challenging environments. We first show that FairDICE can work without reward normalisation, in some cases even when $u_i = \log$. Afterwards, we examine scenarios where FairDICE is trained on data biased to one or specific objective(s), and show that FairDICE finds (relatively) fair policies when given a mixture of biased and unbiased data, but fails to do so when trained on a highly biased dataset. We also examine environments with many (100) rewards and complex (image) observations, and show that FairDICE can scale to these without issue, learning policies which balance all objectives.

### 4.1 Reproductions in discrete environments

### 4.1.1 Learning from a uniform-random policy in MO-Four-Rooms

While verifying Claim 1.1, we were initially unable to reproduce results on MO-FourRooms, as the code for discrete environments was not included in the public repository. Our attempts to recreate the setup using a neural network policy and the described training method yielded significantly worse results than Kim et al. (2025a): at best 10% Util, $-11.3$ NSW and 0.86 Jain, and only after changing the data collection to $\epsilon$-greedy with optimality level 0.2. Key details, such as the state representation and policy parameterisation, were not specified in Kim et al. (2025a), preventing closer replication without access to the original code.

---

[5]Tuned for $\beta \in [10^{-5}, 10^1]$; a sufficiently high $\beta$ would eventually recover the performance of standard Behaviour Cloning.

After obtaining the authors' code for discrete environments, we noted that the original experiments use a tabular policy optimised with MOSEK (ApS, 2025) (which proved essential for good performance), a one-hot state encoding, and a uniformly random data-collection policy with stochasticity of 0.1. Using this setup, we were able to verify Claim 1.1: FairDICE learns a balanced policy which reaches all three goals roughly equally while achieving higher utilitarian welfare than the data-collection policy. We also extended the original experiments with a sweep over $\alpha$ and $\beta$, shown in Fig. 1. The influence of $\beta$ on NSW and Jain's fairness could be seen: lower $\beta$ allows the policy to deviate further from the data, improving fairness when $\alpha > 0$, while higher $\beta$ pushes all configurations toward the behaviour policy. The effect of $\alpha$ is also visible: higher values of $\alpha$ consistently yield higher Jain's fairness at the cost of slightly lower utilitarian welfare, consistent with theoretical predictions.

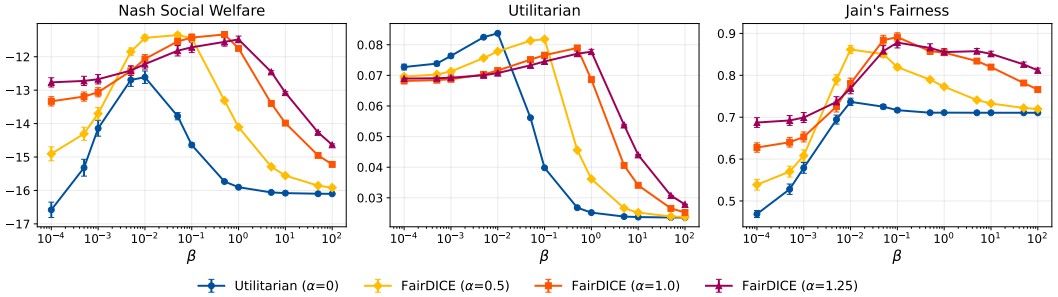

Figure 1: Metrics on MO-FourRooms over a sweep of $\alpha$ and $\beta$ values.

### 4.1.2 Varying hyperparameters in Random MOMDP

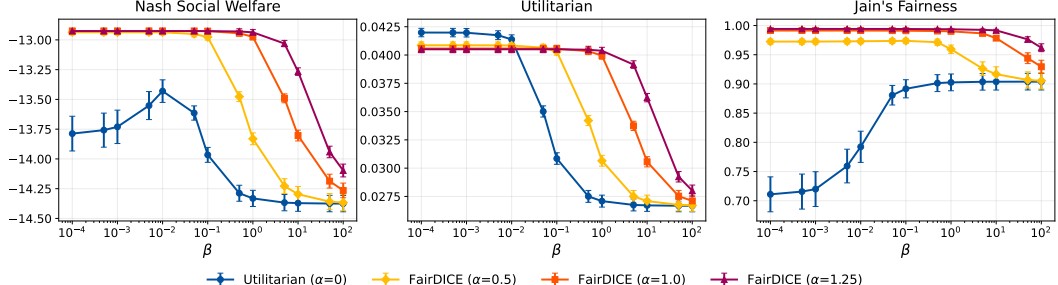

Figure 2: Random MO-MDP metrics over a sweep of $\alpha$ and $\beta$, replicating Fig. 2 from Kim et al. (2025a).

To verify Claim 1.2, we attempt to reproduce the hyperparameter sweep in Random-MOMDP from the original paper. Our results are shown in Fig. 2. When using a tabular policy representation, we observe trends that closely align with Fig. 2 from the original paper (see Appendix F for a direct comparison). FairDICE ($\alpha > 0$) achieves higher NSW and Jain's fairness almost everywhere compared to the utilitarian baseline ($\alpha = 0$), and increasing $\alpha$ yields progressively fairer policies at the cost of utilitarian welfare[6].

The characteristic peak-shaped decline in utilitarian welfare as $\beta$ increases is clearly reproduced, with FairDICE variants maintaining higher welfare longer before dropping. The ordering among FairDICE variants ($\alpha = 1.25 > 1.0 > 0.5$) in both NSW and Jain's fairness is preserved across most $\beta$ values. Notably, at low $\beta$ values, the utilitarian baseline exhibits a transient increase in NSW around $\beta = 10^{-2}$, mirroring a similar pattern in the original results.

Some minor differences remain; utilitarian policies are unable to reach the same level of NSW as FairDICE at any $\beta$, while in the original paper they did at $\beta = 0.01$. Additionally, the utilitarian baseline achieves lower Jain's fairness at small $\beta$ values ($\sim$0.70) compared to the original ($\sim$0.80), suggesting slightly more unequal reward distributions in our randomly generated environments. Despite these differences, the overall conclusions of the original paper regarding the effect of $\alpha$ and $\beta$ are well supported.

---

[6]Except for high $\beta$, where higher $\alpha$ seems to also have a positive impact on all metrics. We suspect (together with the original authors) that this is due to the additional regularisation term being beneficial for learning, but this is uncertain.

### 4.2 Reproductions in continuous environments

### 4.2.1 Consistent performance across values for beta

Kim et al. (2025a) do not specify which $\beta$ they use for the results in continuous environments in the main text. This is somewhat justified by their Appendix I showing stable performance across a wide range of $\beta$, but the previously mentioned broadcasting error suggests that this Claim 2.1 may be misleading. Training on the D4MORL benchmark with code from Kim et al. (2025c), a behavioural cloning baseline and a fixed version of FairDICE with $\lambda \in \{0, 0.0001, 0.1\}$, Fig. 3 shows a selection of the results (see Appendix H for more). In Fig. 3, the original FairDICE implementation matches BC closely, and is indeed not affected[7] by $\beta$. However, the fixed version is highly sensitive to $\beta$ (seemingly less so to $\lambda$), and most values for $\beta$ produce policies worse than standard BC. While some values for $\beta$ can allow FairDICE to slightly outperform the BC baseline, and $\beta \approx 0.1$ often works, there does not appear to be a clear pattern across datasets to how $\beta$ or $\lambda$ should be selected, with many trends not generalising even across these somewhat similar datasets.

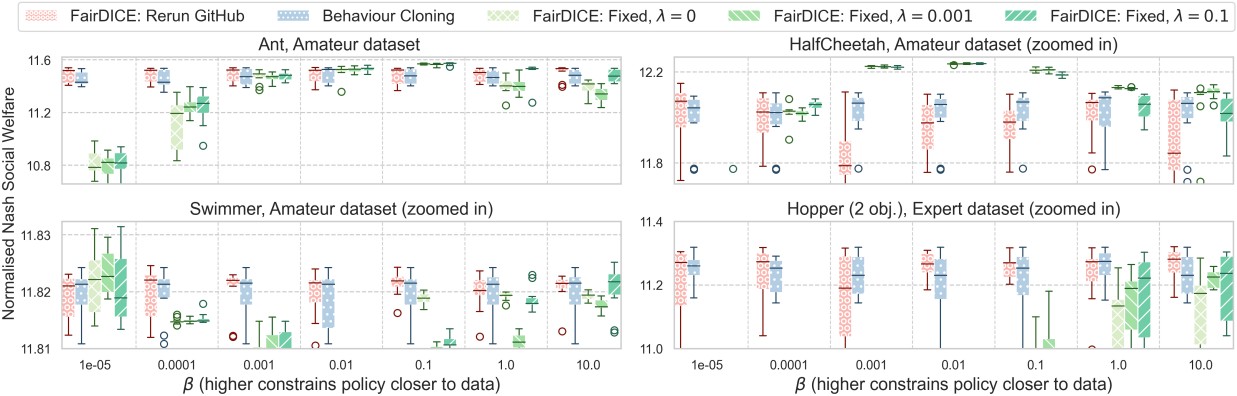

Figure 3: D4MORL NSW boxplots for various losses and $\beta$, using 10 seeds with 100 evaluations per seed.

### 4.2.2 Competitive NSW performance on D4MORL

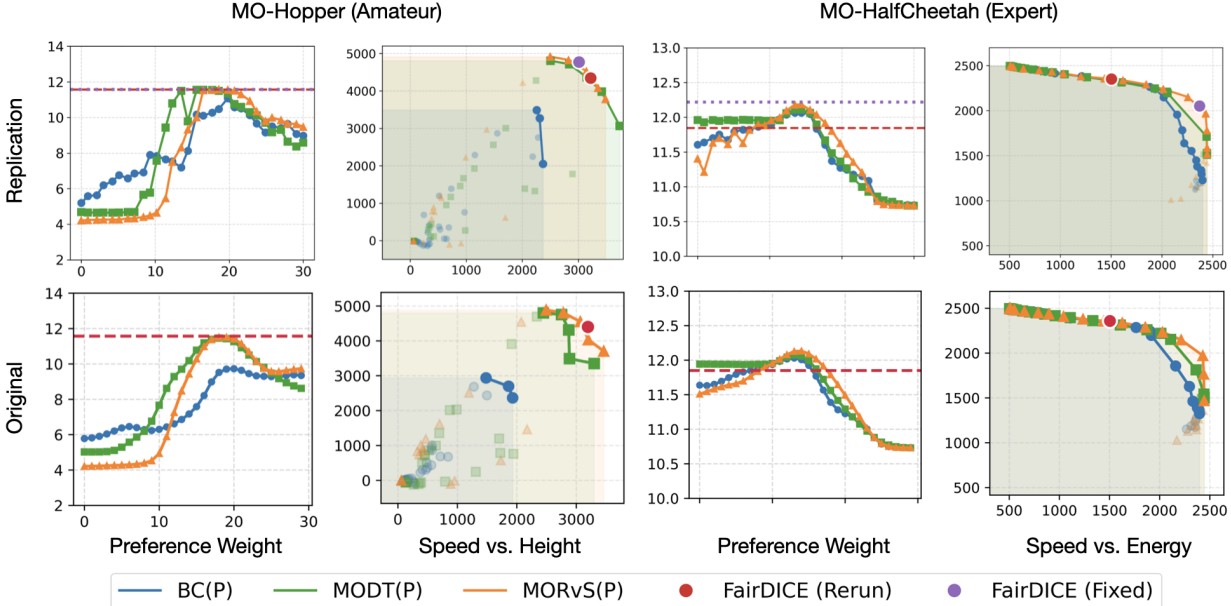

Figure 4: NSW scores and raw return trade-offs with Pareto frontiers on two multi-objective datasets. 'Original' graphs in second row reproduced from Fig. 4 and Fig. 5 of Kim et al. (2025a) with permission.

---

[7]We also performed statistical tests on the impact of $\beta$ in Appendix D which confirm the intuitions given by Fig. 3.

Using the code provided by Zhu et al. (2023) and the same datasets and hyperparameters as in Kim et al. (2025a), we reran all experiments on the D4MORL benchmark. Since the authors do not report the $\beta$ values used in these results, we evaluate FairDICE across a wide range of $\beta$ and select the best-performing setting for each dataset based on NSW. To ensure robust selection, we employ a split-seed protocol: the first 5 seeds are used exclusively for hyperparameter selection, with final results reported on 5 held-out seeds (see Appendix G for the full details of the evaluation protocol).

Qualitatively, the reproduced baseline curves shown in Fig. 4 follow the same trends as those in the original paper (see Appendix G for the full set of results), although they appear noticeably noisier, which we attribute to Kim et al. (2025a) applying a Gaussian filter ($\sigma = 1.2$) to mean performance curves before plotting. We report unsmoothed results to preserve the true variability across preference weights. The "rerun" FairDICE results closely match those reported by Kim et al. (2025a), suggesting at first glance that FairDICE achieves competitive or superior NSW and lies on the Pareto front of the preference-conditioned baselines.

However, as established in Section 3.3.1 and further discussed in Section 4.2.1, the released FairDICE implementation is effectively equivalent to unconditioned behavioural cloning[8] due to a broadcasting error in the policy loss. The strong NSW and Pareto-front performance observed for the "rerun" FairDICE thus likely reflects (1) the correlation of rewards and (2) the tuned BC configuration, rather than active welfare maximisation. Regarding (1), we hypothesise that although D4MORL is a multi-objective dataset, trained agents maximising most intermediate linearisations of the rewards (e.g. [0.4, 0.6] or [0.6, 0.4]) behave similarly enough that a dataset aggregating trajectories from such agents might remain approximately unimodal and suitable from direct BC. For (2), the comparison must be made with the BC baseline from Zhu et al. (2023), which did not achieve Pareto-optimality; the difference here lies primarily in training tricks[9] used to boost performance. Additionally, it should be noted that the "rerun" FairDICE outperforms amateur training datasets it learns from, seeming due to learning to average out the noise in the amateur dataset[10].

Examining the effect of FairDICE weights in weighted BC, we also evaluate the "fixed" FairDICE implementation. Its performance is comparable to BC on most datasets, with one notable exception: on MO-HalfCheetah (Expert), shown on the right of Fig. 4, it outperforms the "rerun" version significantly in NSW while lying on the Pareto front. Overall, while our reproduction matches the qualitative patterns reported in the original paper, Claim 2.2 is not robustly supported, as the contribution of FairDICE's welfare mechanism to the observed NSW remains unclear and may be attributable to the dataset or improved architecture.

### 4.3 Extensions to the original paper

Sec. 4.1 showed that many of the claims regarding theoretical properties and performance of FairDICE hold in discrete environments, but Sec. 4.2 showed that (beyond the error in code) that FairDICE often performs similarly to standard BC when $\beta$ is tuned, and worse for many choices of $\beta$. To better characterise the scenarios in which FairDICE might prove effective, we provide four extensions to the original paper: Sec. 4.3.1 and Sec. 4.3.2 investigate possible failure modes in negative rewards and unbalanced data, while Sec. 4.3.3 and Sec. 4.3.4 investigate the performance of FairDICE in complex environments beyond D4MORL.

### 4.3.1 Handling negative returns without normalisation

Using $u_i = \log$ to incentivise fairness amongst objectives might appear problematic with negative returns, but in a reviewer rebuttal, Kim et al. (2025b) make the claim that extending FairDICE to handle negative returns directly is possible by using a piecewise alternative to the logarithm as $u_i$, defined as $g(x) = \log(x)$ when $x \geq 1$ and $g(x) = -0.5(x - 2)^2 + 0.5$ when $x < 1$. Ext. 3.1 of FairDICE therefore investigates the practical performance of this $g$ by training on Hopper and Walker2d (which have negative rewards) using standard (fixed) FairDICE, log-based FairDICE without normalisation and the piecewise-log $u_i(x) = g(x)$.

---

[8]Note that the baseline method BC(P) uses preference conditioning and a slightly different architecture (see Zhu et al., 2023), which explains why BC(P) and FairDICE (Rerun) differ in NSW despite both performing behavioural cloning. As explained in the rest of the paragraph, the unconditioned BC baseline from Zhu et al. (2023) underperforms due to missing training tricks.

[9]Kim et al. (2025a) use a cosine learning-rate schedule (Loshchilov & Hutter, 2017) and orthogonal initialisation (Saxe et al., 2014a) for the weight matrices in the MLP. Both are necessary to obtain the high performance observed in Fig. 4.

[10]To understand this, we must note that although Kim et al. (2025a) describe their model as a Gaussian policy, during evaluation the mean of the action distribution is used instead of sampling. Since the amateur dataset is only suboptimal in that it contains white-noised versions of expert actions, the mean of the learned Gaussian can approximate the true expert action. If we instead *sample* an action from the predicted distribution (as in discrete environments), performance degrades significantly.

Fig. 5 shows that FairDICE is able to achieve comparable performance with the adjusted $u_i$ as with log. However, even with $u_i = \log$, FairDICE is able to perform well on unnormalised rewards, because (unlike the original objective) the FairDICE loss only uses $u_i$ in the regularisation term for $\mu$. As such, log-FairDICE can handle occasional negative rewards or returns, so long as the *expected* return[11] remains positive.

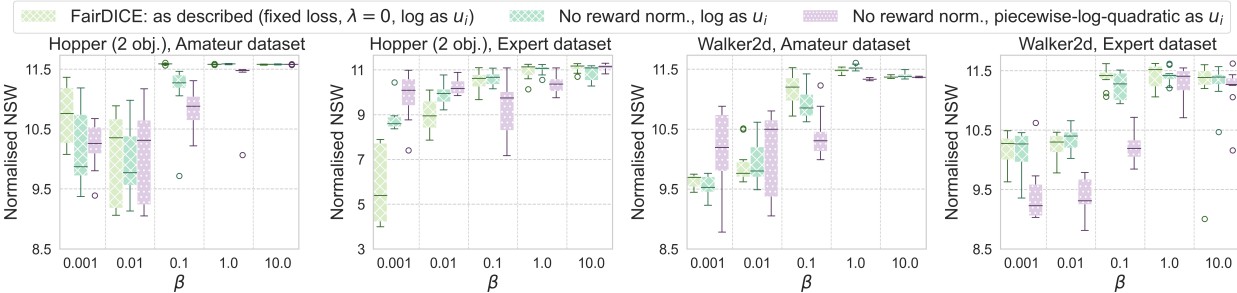

Figure 5: FairDICE performance with/without reward normalisation and with piecewise-log for $u_i$, trained on Hopper and Walker2d from D4MORL. Boxplots are drawn using 10 seeds with 100 evaluations per seed.

### 4.3.2 Learning from policies biased to specific rewards

In the same reviewer rebuttal discussed in Section 4.3.1, Kim et al. (2025b) claim that FairDICE is robust to suboptimal data quality and limited coverage. To examine this claim, we probe FairDICE's robustness to biased datasets in a discrete environment, namely a biased MO-FourRooms dataset where visits to the 3 goals were split 80/10/10%. This checks if FairDICE can achieve fairness despite an unfair offline dataset.

As shown in Fig. 6, FairDICE with $\alpha = 1.0$ on the imbalanced dataset (red) substantially outperforms the utilitarian baseline on imbalanced data (which inherits the dataset's bias), demonstrating that the fairness mechanism provides meaningful correction. However, the imbalanced case does not fully recover the performance of the balanced setting: at low $\beta$, NSW is approximately 10 points lower, and Jain's fairness peaks around 0.7–0.8 compared to $\sim 0.9$ for the balanced dataset, with considerably wider confidence intervals. The two cases converge only at high $\beta$, where the policy is constrained close to the data regardless of $\alpha$. This suggests that FairDICE can partially mitigate dataset imbalance, but does not fully overcome it in discrete environments.

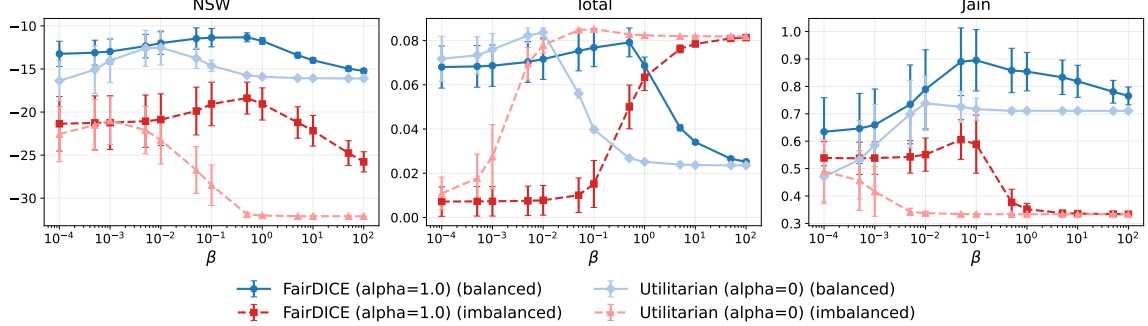

Figure 6: Metrics on MO-FourRooms for policies trained on balanced and imbalanced datasets

We also examine the performance of FairDICE on D4MORL tasks with adjusted distributions over preference weights in Appendix K, using the pre-defined[12] alternate folds ('narrow' and 'wide', as defined by Zhu et al., 2023), but find very little difference in this case. We hypothesise that this may be, as mentioned previously, due to that many of the trained policies used to generate the D4MORL dataset behave similarly.

---

[11]Kim et al. (2025a) make use of slack variables $k_i$ in their derivation of FairDICE, where each $k_i$ represents the expected return for objective $i$. The regularisation term in FairDICE (which is the only term containing $u_i/\log$) contains the term $u_i(k_i)$, which must be valid for the FairDICE objective to be well-defined (for details, see Kim et al., 2025a). As such, individual rewards or returns can be out-of-domain for the chosen $u_i$, so long as the expected return $k_i$ is a valid argument to $u_i$. It should be noted that the nature of gradient-based optimisers makes it unlikely that $k_i$ would actually become negative, but the weights $w^*$ in a scenario with negative expected return may not be meaningful, or defined according to the FairDICE formulation.

[12]For completeness we also provide our replication of Table 4 from Kim et al. (2025a) in Appendix J using the GitHub implementation (BC) which uses non-pre-defined folds, but we do not have the corresponding results for "fixed" FairDICE.

### 4.3.3 Learning with high-dimensional rewards

Kim et al. (2025a) make the claim that FairDICE can efficiently scale to environments with high-dimensional rewards. In terms of efficiency, FairDICE requires only one additional parameter per reward, but Ext. 3.3 examines whether learned policies remain effective in an example high-dimensional reward environment named GroupFair. In this environment with 100 rewards, repeated unfair actions lead to progressively more unfair outcomes, while random actions eventually do as well; high NSW therefore requires a fairness-aware policy. Fig. 7 shows that FairDICE was able to approach such a fairness-aware policy when given training data from a random policy, but failed to do so when data came from a biased policy. Hyperparameter $\lambda$ did not have any noticeable effect and is therefore omitted from Fig. 7, while $\beta$ does seem to have some effect on performance in GroupFair (though the pattern is unclear).

### 4.3.4 Scaling to a complex environment

To verify Ext. 3.4, we choose to train on an environment with large state space: each observation is a rendered image from a minecart simulator, significantly more complex than previous environments.

| $\alpha\backslash\beta$ | 0.001 | 0.01 | 0.1 | 1.0 | 10.0 | 100.0 |
|---|---|---|---|---|---|---|
| 0.0 | $3.92 \pm 0.12$ | $3.92 \pm 0.06$ | $3.97 \pm 0.23$ | $4.08 \pm 0.09$ | $3.96 \pm 0.06$ | $3.93 \pm 0.15$ |
| 1.0 | $4.09 \pm 0.00$ | $4.12 \pm 0.11$ | $4.23 \pm 0.11$ | $4.20 \pm 0.06$ | $3.99 \pm 0.03$ | $4.05 \pm 0.06$ |

Table 1: NSW on MO-Minecart-RGB across a sweep of $\beta$ on utilitarian and $\alpha = 1$, where the training dataset had a NSW of $1.24 \pm 3.784$. The results are averaged over 3 seeds

Table 1 presents NSW scores across a sweep of $\beta$ for both utilitarian ($\alpha = 0$) and fair ($\alpha = 1$) objectives. FairDICE substantially outperforms the data-collection policy (NSW $1.24 \pm 3.78$), achieving scores around 4.0 across all tested configurations. The fair objective ($\alpha = 1.0$) consistently yields slightly higher NSW than the utilitarian baseline, with the best performance of $4.23 \pm 0.11$ at $\beta = 0.1$. NSW remains relatively stable across the tested range of $\beta \in \{0.001, 100\}$, suggesting that FairDICE is less sensitive to hyperparameter choices in this particular environment compared to the D4MORL tasks.

These results provide preliminary evidence that FairDICE can scale to image-based observations, though the improvement margin over the utilitarian baseline is small. The high variance in the dataset NSW (due to the expert policy's variable performance across objectives) makes direct comparison challenging. Future work could investigate whether the relative stability across $\beta$ is a property of image-based environments, or whether differences between values of $\beta$ will become more distinct after running on more seeds. Fig. 8 shows pairwise objective trade-offs for MO-Minecart-RGB across $\beta$ values.

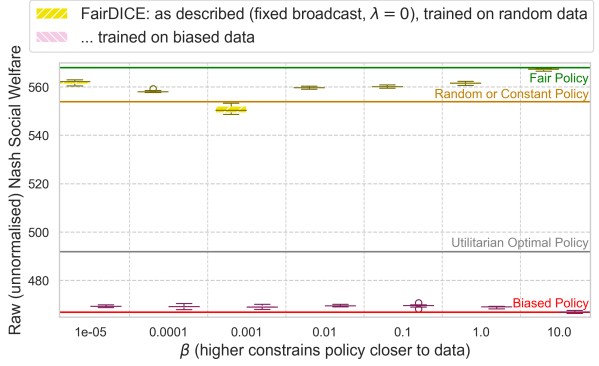

Figure 7: FairDICE performance on GroupFair environment when trained on biased/random data, 10 seeds with 100 evaluations per seed. Lines indicate the performance of policies from Appendix B.4.

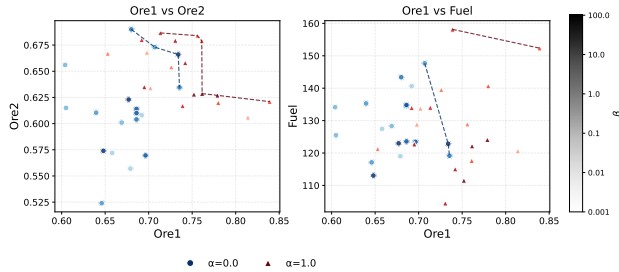

Figure 8: MO-Minecart-RGB results (pareto fronts estimated on betas)

# 5 Discussion

Kim et al. (2025a) propose an extension to OptiDICE which can learn a fair compromise between objectives automatically, and we replicate in Sec. 4.1.1 and 4.1.2 that these properties can be effective in improving NSW in toy environments, as well as in Sec. 4.2.2 for continuous environments. However, the original results given for continuous environments are incorrect, and Sec. 4.1.2 and Sec. 4.2.1 show that the actual performance of FairDICE is highly dependent on the choice of $\beta$. While this is common in many offline RL algorithms (Kumar et al., 2022), it goes against Claim 2.1 and shows that applying FairDICE reliably requires, at a minimum, evaluating a variety of $\beta$ (which is challenging in truly offline settings). Besides the corrections to performance estimates of FairDICE, Sec. 4.2 also shows that BC, if done well, can yield Pareto-efficient policies on many D4MORL datasets, including supposedly amateur ones. This suggests that future work may wish to use more challenging datasets to better display the benefits of offline RL. Two candidates for such challenging environments are used in Sec. 4.3.3 and 4.3.4, which show that FairDICE can effectively scale to many rewards or complex environments — when a search through possible values for linearisation weights $\mu$ might not suffice. This result is made somewhat weaker, however, by Sec. 4.3.2 (and in part Sec. 4.2.2) suggesting the essential role of a balanced data-collection policy in NSW performance. In conclusion, the theoretical approach taken by Kim et al. (2025a) seems well-supported, but the situations in which FairDICE can easily be applied for good performance are more limited than initially presented.

Future work could investigate the application of a similarly learnable linearisation to other offline RL algorithms (e.g. Kostrikov et al., 2021; Kumar et al., 2020) to see if this yields stabler results than with DICE. Considering the novel mechanism (learning $\mu$ via a regularisation term) does not appear to introduce more tuning, if this approach could be combined with an offline RL framework more robust to hyperparameter selection, the result might be usable in truly offline settings. More generally, investigating, comparing or proposing alternate methods for supporting non-linear utilities (e.g. Agarwal et al., 2022; Park et al., 2024) could improve or clarify existing options for fair (offline) MORL. Many methods (e.g. Park et al., 2024) exist for performing RL with non-linear scalarisation in the online case, and it may be valuable to investigate how these methods can be applied in an offline setting.

## 5.1 Challenges during reproduction

Much of Kim et al. (2025a) was difficult to reproduce exactly, with most challenges stemming from two sources. Firstly, the provided code seemed to use Jax 0.4 and similarly deprecated versions for other libraries, which caused compatibility issues on modern hardware. We attempted to rewrite FairDICE into PyTorch 2.9, but could not exactly replicate the performance of the original code and had to discard this effort. Secondly, as mentioned previously, many of the important hyperparameters (e.g. model architecture or optimiser) were unclear for the experiments in discrete environments, and our guesswork could not fill the gaps to a perfect replication. Later correspondence with the original authors provided the necessary details, but also caused additional delays. To aid scientific reproducibility, we would advise future work to write experiments using latest library versions (or release a list of package versions used), and release all code used for experiments.

## 5.2 Computational requirements and environmental impact

Experiments were run across four machines, using a combined 760 hours of GPU-compute. During experiments, GPU VRAM usage was below 20GB (except Minecart-RGB, which used 80GB). As a consequence of the electricity used during experiments, we estimate to have generated approximately 95 kg of CO2eq. More details on the hardware and environmental impact of this replication study are in Appendix I.

## 5.3 Correspondence with original authors

In mid January, we sent an e-mail to the corresponding authors of Kim et al. (2025a) with questions about the implementation of the discrete experiments and the interpretation of their results. In early February, we received a reply, after which we exchanged several e-mails regarding the discrepancies in the public implementation of the experiments in continuous environments. In late February, the issues with continuous environments were confirmed, and it became clear that the original authors sought to revise the existing code (though this has not yet been done as of early May). In early March, we requested and received feedback on the final draft of this report.

## Acknowledgments

We would like to thank our supervisor Satchit Chatterji for their valuable feedback on our process and earlier revisions of this report. We would also like to thank the University of Amsterdam for providing computational resources (the first machine in Appendix I) for this project. Finally, we would like to thank Woosung Kim for answering our many questions regarding the implementation and theory of FairDICE, and for helping reformulate some parts of our Background section discussing FairDICE.

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

# A    Fair MORL via Stationary Distribution Correction

FairDICE, as a DICE-based method (Nachum et al., 2019), is derived from correction estimates of the stationary visitation distribution $d(s, a)$. This stationary visitation distribution can be seen as a weighting over states, indicating which states are relevant for the discounted return. For details, see Kim et al. (2025a).

FairDICE proposes to learn a critic $\nu(s)$ and a preference vector $\mu$ using the following loss:

$$\mathcal{L}(\nu, \mu) = \mathbb{E}_{\dot{s} \sim p_0} \left[ (1 - \gamma)\nu(\dot{s}) \right] + \mathbb{E}_{(s,a) \sim \mathcal{D}} \left[ w^*(s,a)e(s,a) - \beta f(w(s,a)) \right] + \sum_i \left( u_i(k^*) - \mu_i k^* \right). \tag{2}$$

However, evaluating $e(s, a)$ as described in Kim et al. (2025a) requires transition probabilities $T$, which are typically unknown in (offline) RL. In their code (Kim et al., 2025c), we instead find a TD-style update:

$$\mathcal{L}(\nu, \mu) = \mathbb{E}_{\dot{s} \sim p_0} \left[ (1 - \gamma)\nu(\dot{s}) \right] + \mathbb{E}_{(s,a,r,s') \sim \mathcal{D}} \left[ w^*(s,a,r,s')e(s,a,r,s') - \beta f(w(s,a,r,s')) \right] + \sum_i \left( u_i(k^*) - \mu_i k^* \right), \tag{3}$$

$$e(s, a, r, s') = \sum_i \mu_i r_i + \gamma \nu(s') - \nu(s), \tag{4}$$

$$w^*(s, a, r, s') = \max\{0, (f')^{-1}(\frac{1}{\beta} e(s, a, r, s'))\}, \tag{5}$$

$$k_i^* = (u_i')^{-1}(\mu_i), \tag{6}$$

$$\mathcal{L}(\pi) = -\mathbb{E}_{(s,a,r,s') \sim \mathcal{D}} \left[ w^*(s, a, r, s') \log \pi'(a \mid s) \right], \tag{7}$$

where $f$ is a divergence (set to the Soft-$\chi^2$ divergence for all experiments) and $u_i$ is a non-linear transformation of the $i$'th reward (often set to log for all $i$, such that $\sum_i u_i(J(\pi))$ corresponds with NSW). The policy (e.g. an MLP) is extracted using weighted behaviour cloning, where $w^*(s, a, s')$ is normalised to have a mean of 1 across each batch for the policy loss $\mathcal{L}(\pi)$.

For more details on DICE-based methods, we direct the reader to the tutorial paper Nachum & Dai (2020) as well as Mao et al. (2024) for a more thorough overview of the related theory, while Kim et al. (2025a) explain FairDICE in some detail in Sec. 3, 4, 5 and Appendix A.

# B    Detailed environment descriptions

## B.1    MO-Four-Rooms

This environment has four rooms and three goals, with one-hot rewards granted for reaching each of the three goals. Following the authors recipe, we generated the datasets using a uniformly random policy. For simple discrete environments policy is represented in tabular form and optimized with MOSEK (ApS, 2025). A sample render of the environment can be seen on Fig. 9

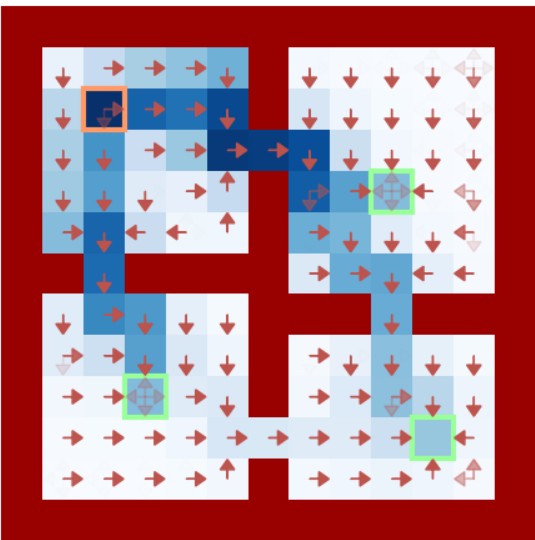

Figure 9: Visualization of the MO-FourRooms environment together with a sample policy's state visitation heatmap

## B.2    Random MOMDP

The environment parameters, such as number of states, actions, and goal is identical to the ones described in the original paper. Following the author's setup for MO-FourRooms, we learn a tabular policy with MOSEK solver. All results are averaged over 1000 seeds.

Again indentically to the original paper, we collect a dataset of 100 trajectories under a behavior policy with optimality level 0.5: the data collection policy selects the optimal action with probability 0.5 and a uniformly random action otherwise.

### B.3 Multi-Objective MuJoCo environments based on D4RL benchmark

The original D4RL benchmark from Fu et al. (2020) was augmented in Zhu et al. (2023) to include five 2-objective environments and one 3-objective environment, and is the primary method used by Kim et al. (2025a) for evaluating FairDICE in a continuous environment. The name of the new multi-objective variant of D4RL is D4MORL. All environments feature 8-27 continuous inputs and 2-8 continuous outputs (actions), with rewards being given for moving 'forward' and minimising the norm of actions taken (and for staying vertically high, in the three-objective variant of Hopper). More details on the specific reward formulae can be found in Kim et al. (2025a) or Zhu et al. (2023).

All models used an MLP for both the critic $\nu$ and the policy $\pi$. For most environments, these had 3 layers of 768 hidden units, with two exceptions: Ant had only 512 hidden units, while the three-objective variant of Hopper used four layers (of 768). Policies parametrised the mean and standard deviation of (independent) Gaussian distributions for each action.

For both training datasets and evaluation rollouts, environments truncate after 500 timesteps (though some environments such as Hopper also have early termination conditions)

### B.3.1 Datasets in D4MORL

Each dataset is collected using expert policies (which match the target preferences) and amateur policies (a mix of the expert policy and stochastic actions), where amateur policies create a more diverse dataset. A more detailed description of the dataset collection procedure can be found in Kim et al. (2025a) and Zhu et al. (2023). The distribution of the target preferences of each dataset can be found in Fig. 10, which illustrates the differences between the distribution types; uniform, narrow, and wide. It can be noted that the different distributions are not uniform among all environments. This is a result of a differing preference space for each environment, as some combinations of preference weights are not achievable. This can be clarified further using an example from Zhu et al. (2023), where MO-Hopper cannot have a 100% preference for running (the first objective), as jumping (the second objective) is necessary for gaining running rewards.

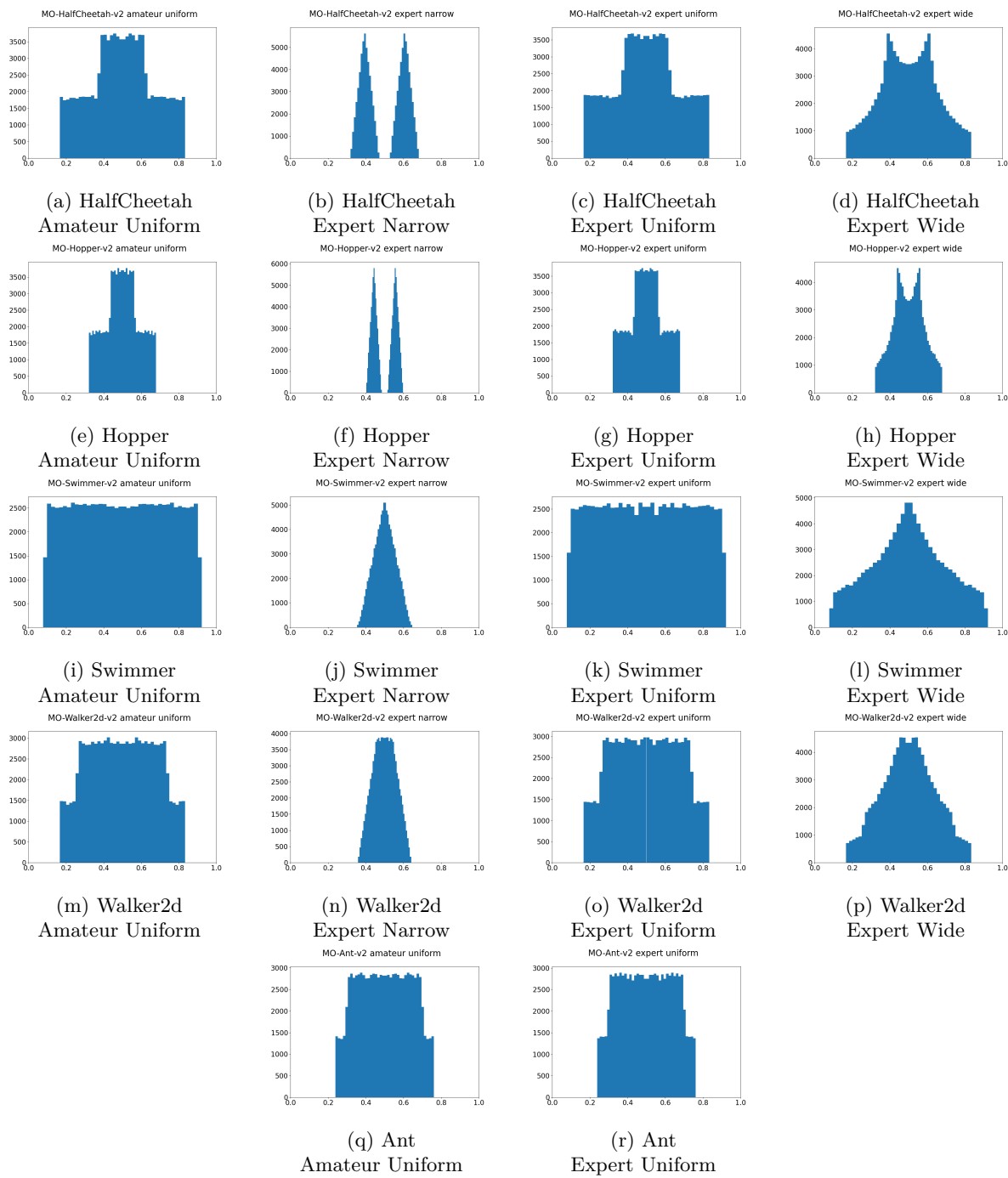

Figure 10: Distribution of (individual) preference weights in all two-objective environments in D4MORL, divided over 40 bins.

### B.4   MO-GroupFair

As part of Ext. 3.3, we define a novel environment representing a miniature model of social unfairness and compounding inequality. In **MO-GroupFair**, there exist 5 groups $\mathcal{G}_1, \ldots \mathcal{G}_5$ and 100 individuals, where each individual has three group memberships. The agent (e.g. a government) has 1 unit of reward each timestep and can distribute this according to the 7 random options $O = \mathbb{R}^{7 \times 5}$. One such option might be $O_1 = [0.5, 0.1, 0.1, 0.1, 0.2]$, where the first group receives more of the reward than the rest. Upon selecting an option $a$, each individual $x$ has a utility equal to the sum of the reward which their groups obtained: $r_x = \sum_{i \mid x \in \mathcal{G}_i} O_{a,i}$. Afterwards, the distribution from which $O$ is drawn is adjusted, such that groups which received more than the average reward this timestep are more likely to have options favouring them in future timesteps, and vice versa.

Group memberships are uniformly random for each individual, and an individual can be part of a group multiple times (interpreted as having a stronger affiliation for the group). $O$ is composed of 7 i.i.d. draws from a Dirichlet distribution, where the concentration parameters are initially uniform: $\alpha = \mathbf{1}$. For each group, a running total of obtained rewards is kept: $g_{i,T} = \sum_{t=1}^{T} O_{a_t,i}^{(t)}$ where $O^{(t)}$ and $a_t$ are the corresponding values at timestep $t$. The advantage of a group is then $\beta_{i,T} = \tanh(\frac{1}{10}(g_{i,T} - \bar{g}_{:,T}))$, where $\bar{g}_{:,T}$ is the mean total reward across groups at timestep $T$, using which we define $O^{(T)} \sim \text{Dirichlet}(\alpha + \beta_{:,T})$. At each timestep, the policy observes the entirety of $O$, and must output a discrete action as its choice amongst the 7 options.

While the above description allows for multiple different configurations in terms of group memberships, experiments for Ext. 3.3 fixed the memberships to one specific configuration, with group sizes of $[69, 46, 63, 74, 48]$ (generated using seed 42).

For this environment, four reference policies are used (shown as lines in Fig. 7):

1. Random: A random policy (equivalent in performance to a constant policy).
2. Biased: A policy which always selects the option most favourable to the first group.
3. Util. Optim.: A policy which always selects the option most favourable to the largest (fourth) group. This rapidly results in rewards being concentrated on this group, which means few rewards are 'wasted' on groups with less membership and thereby a lower reward multiplier.
4. Fair: A policy which selects the option with the highest NSW amongst groups (not individuals). This is not an optimal policy w.r.t. NSW.

Offline RL datasets are generated using the first and second reference policies, by running 5000 rollouts and gathering the results. All rollouts (including evaluation rollouts) terminate after 500 timesteps.

### B.5   MO-Minecart-RGB

For Ext. 3.4, we used an environment where a cart moves across a rectangular world and mines two types of ore.

We used a CNN encoder with 5 layers, each applying a $(3 \times 3)$ kernel. The number of feature maps across layers was $[1, 32, 64, 64, 64, 64]$, where the first channel corresponds to the grayscale input image.

To create the dataset, we trained several policies using PPO with different reward scalarization weights: one set of weights ignored the fuel component entirely, while the others included it with small weights. The fuel component of the reward vector originally ranged from $-1$ to $0$, so we added 1 at each step to ensure the non-negativity assumption from the original paper holds.

## C   Training hyperparameters

All models except for tabular policies are trained using Adam with a learning rate of 0.0003 and no weight decay. For $\mu$ and $\nu$, the learning rate is held constant, but for the policy $\pi$ a cosine schedule is used which decays to 0 at the end of training. Models are trained for 10000 iterations with a batch size of 256 (approximately one pass over the D4MORL datasets) except for Minecart-RGB, which uses a batch size of 64. In the models, biases are zero-initialised, while the weight matrices of affine ('Linear') layers are orthogonally initialised with gain $\sqrt{2}$ (Saxe et al., 2014b). The exception to this is the affine layers leading to standard deviation outputs for Gaussian policies, which are initialised with a gain of 0.001 instead. Tabular policies are optimized with 'cvxpy' (Diamond & Boyd, 2016; Agrawal et al., 2018) and a MOSEK solver, the results are averaged over 1000 seeds.

In terms of normalisation, Kim et al. (2025a) normalise continuous states using precomputed means and standard deviations taken from Zhu et al. (2023), by subtracting the mean of each state dimension and dividing by the standard deviation. Rewards are normalised by default (to prevent negative expected returns) by determining the minimum and maximum reward for each objective in the dataset (different between policy sources, e.g. expert v.s. amateur in D4MORL) and performing min-max normalisation:

$$r^{norm} = \frac{r - r_{min}}{r_{max} - r_{min}}$$

As mentioned in Sec. 3, this is also done during evaluation to obtain the 'Normalised Nash Social Welfare'. All of the above mirrors the training code from Kim et al. (2025c).

# D  Statistical testing

```
Group (Env/Type)                | H-Statistic  | P-Value    | Significance
------------------------------------------------------------------------------------
Ant-v2/amateur                  | 2.7228       | 0.8427     | Not Significant
Ant-v2/expert                   | 1.9062       | 0.9281     | Not Significant
HalfCheetah-v2/amateur          | 8.3690       | 0.2123     | Not Significant
HalfCheetah-v2/expert           | 4.4256       | 0.6193     | Not Significant
Hopper-v2/amateur               | 2.0578       | 0.9143     | Not Significant
Hopper-v2/expert                | 3.2583       | 0.7758     | Not Significant
Hopper-v3/amateur               | 8.0930       | 0.2314     | Not Significant
Hopper-v3/expert                | 5.9783       | 0.4256     | Not Significant
Swimmer-v2/amateur              | 2.1623       | 0.9042     | Not Significant
Swimmer-v2/expert               | 4.5984       | 0.5962     | Not Significant
Walker2d-v2/amateur             | 4.1800       | 0.6523     | Not Significant
Walker2d-v2/expert              | 9.1234       | 0.1668     | Not Significant

Group (Env/Type/Grad)                    | H-Statistic  | P-Value    | Significance
------------------------------------------------------------------------------------------
Ant-v2/amateur/Fixed-Grad0               | 60.8201      | < 0.001    | SIGNIFICANT
Ant-v2/amateur/Fixed-Grad0.0001          | 59.8181      | < 0.001    | SIGNIFICANT
Ant-v2/amateur/Fixed-Grad0.1             | 63.2856      | < 0.001    | SIGNIFICANT
Ant-v2/expert/Fixed-Grad0                | 47.1837      | < 0.001    | SIGNIFICANT
Ant-v2/expert/Fixed-Grad0.0001           | 52.7054      | < 0.001    | SIGNIFICANT
Ant-v2/expert/Fixed-Grad0.1              | 35.6895      | < 0.001    | SIGNIFICANT
HalfCheetah-v2/amateur/Fixed-Grad0       | 66.8894      | < 0.001    | SIGNIFICANT
HalfCheetah-v2/amateur/Fixed-Grad0.0001  | 62.0095      | < 0.001    | SIGNIFICANT
HalfCheetah-v2/amateur/Fixed-Grad0.1     | 66.4743      | < 0.001    | SIGNIFICANT
HalfCheetah-v2/expert/Fixed-Grad0        | 56.8699      | < 0.001    | SIGNIFICANT
HalfCheetah-v2/expert/Fixed-Grad0.0001   | 53.0579      | < 0.001    | SIGNIFICANT
HalfCheetah-v2/expert/Fixed-Grad0.1      | 57.2693      | < 0.001    | SIGNIFICANT
Hopper-v2/amateur/Fixed-Grad0            | 61.4923      | < 0.001    | SIGNIFICANT
Hopper-v2/amateur/Fixed-Grad0.0001       | 57.7788      | < 0.001    | SIGNIFICANT
Hopper-v2/amateur/Fixed-Grad0.1          | 60.3090      | < 0.001    | SIGNIFICANT
Hopper-v2/expert/Fixed-Grad0             | 60.6420      | < 0.001    | SIGNIFICANT
Hopper-v2/expert/Fixed-Grad0.0001        | 57.1366      | < 0.001    | SIGNIFICANT
Hopper-v2/expert/Fixed-Grad0.1           | 60.1912      | < 0.001    | SIGNIFICANT
Hopper-v3/amateur/Fixed-Grad0            | 62.4116      | < 0.001    | SIGNIFICANT
Hopper-v3/amateur/Fixed-Grad0.0001       | 55.6563      | < 0.001    | SIGNIFICANT
Hopper-v3/amateur/Fixed-Grad0.1          | 59.8223      | < 0.001    | SIGNIFICANT
Hopper-v3/expert/Fixed-Grad0             | 36.6881      | < 0.001    | SIGNIFICANT
Hopper-v3/expert/Fixed-Grad0.0001        | 46.7355      | < 0.001    | SIGNIFICANT
Hopper-v3/expert/Fixed-Grad0.1           | 37.5463      | < 0.001    | SIGNIFICANT
Swimmer-v2/amateur/Fixed-Grad0           | 52.9202      | < 0.001    | SIGNIFICANT
Swimmer-v2/amateur/Fixed-Grad0.0001      | 54.0313      | < 0.001    | SIGNIFICANT
Swimmer-v2/amateur/Fixed-Grad0.1         | 61.6752      | < 0.001    | SIGNIFICANT
Swimmer-v2/expert/Fixed-Grad0            | 41.0522      | < 0.001    | SIGNIFICANT
Swimmer-v2/expert/Fixed-Grad0.0001       | 15.3640      | 0.0176     | SIGNIFICANT
Swimmer-v2/expert/Fixed-Grad0.1          | 26.4109      | < 0.001    | SIGNIFICANT
Walker2d-v2/amateur/Fixed-Grad0          | 56.1406      | < 0.001    | SIGNIFICANT
Walker2d-v2/amateur/Fixed-Grad0.0001     | 57.0340      | < 0.001    | SIGNIFICANT
Walker2d-v2/amateur/Fixed-Grad0.1        | 57.1514      | < 0.001    | SIGNIFICANT
Walker2d-v2/expert/Fixed-Grad0           | 57.2369      | < 0.001    | SIGNIFICANT
Walker2d-v2/expert/Fixed-Grad0.0001      | 50.8998      | < 0.001    | SIGNIFICANT
Walker2d-v2/expert/Fixed-Grad0.1         | 53.6441      | < 0.001    | SIGNIFICANT
```

Figure 11: FairDICE statistical tests, comparing the original (top) and corrected (bottom) implementations.

To analyze the impact of the hyperparameter $\beta$, we aggregated performance scores across six environments and two dataset qualities. For each condition, we grouped results by $\beta$ (comparing 7 values from $10^{-5}$ to 10), with each group consisting of 10 independent seed evaluations. For the corrected implementation, we further stratified this analysis by the gradient penalty coefficient $\lambda \in 0, 10^{-4}, 10^{-1}$.

We employed the Kruskal-Wallis H-test to assess significance. This non-parametric test is appropriate here as it allows for the comparison of multiple independent groups simultaneously without assuming the data follows a normal distribution, which is crucial given the sample size ($N = 10$).

As shown in Fig. 11, the results establish a clear contrast. For the original implementation (top), the test yielded no statistically significant differences ($p > 0.05$) across any environment, confirming that the algorithm was effectively insensitive to $\beta$. Conversely, for the corrected implementation (bottom), highly significant differences ($p < 0.001$) were observed across nearly all cases, regardless of the gradient penalty $\lambda$ applied. This statistically confirms that the stability observed in the original work was an artifact of the implementation error, and that the corrected algorithm is, in fact, highly sensitive to $\beta$.

## E   Baseline details

For all baselines, we train on 5 random seeds for each environment and dataset configuration, with 10 evaluation episodes per seed. The hyperparameters used for the baselines are summarized in Table 2. A granularity of 29 divides the range $[0, 1]$ into 29 intervals, resulting in 30 evaluations.

| Hyperparameter | BC(P) | MODT(P) | MORvS(P) |
|---|---|---|---|
| Batch Size | 256 | 256 | 256 |
| Training Steps | 100,000 | 100,000 | 100,000 |
| Optimizer | Adam | Adam | Adam |
| Normalize Reward | True | True | True |
| Condition on Preference ($\mu$) | ✓ | ✓ | ✓ |
| Condition on Return-to-Go | × | ✓ (Multi-obj) | ✓ (Multi-obj) |
| Condition on State | ✓ | ✓ | ✓ |
| Concatenate $\mu$ to State | ✓ | ✓ | ✓ |
| Concatenate $\mu$ to RTG | N/A | ✓ | No |
| Concatenate $\mu$ to Action | N/A | ✓ | No |
| Granularity | 29 | 29 | 29 |
| Hidden Size / Embed Dim | 512 | 512 | 512 |
| Number of Layers | 3 | 3 | 3 |
| Attention Heads | N/A | 1 | N/A |

Table 2: Hyperparameters for BC(P), MODT(P), and MORvS(P) baselines.

## F Random MOMDP Figure Comparison

In Sec. 4.1.2 we replicate Fig. 2 from Kim et al. (2025a). For ease of comparison, we show both the original graph in Fig. 12 and the replication in Fig. 13 together below. The general trends remain the same, but some minor details and specific values differ (likely due to subtle differences in how the MOMDP is generated).

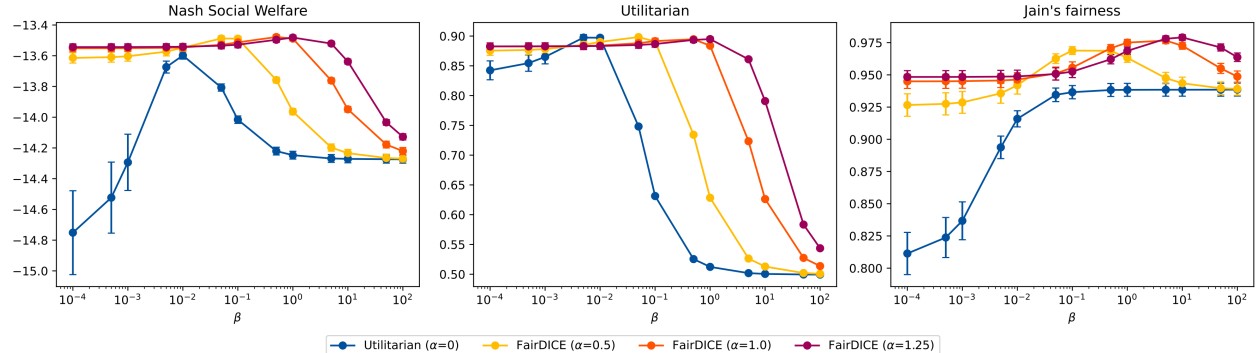

Figure 12: Original Fig. 2 from Kim et al. (2025a). Original caption: "Policy performance on Random MOMDP domain across different $\alpha$ and $\beta$ values, evaluated on Nash social welfare, Utilitarian welfare, and Jain's fairness index. Results are averaged over 1000 seeds, and reported with 95% confidence intervals". Reproduced with permission.

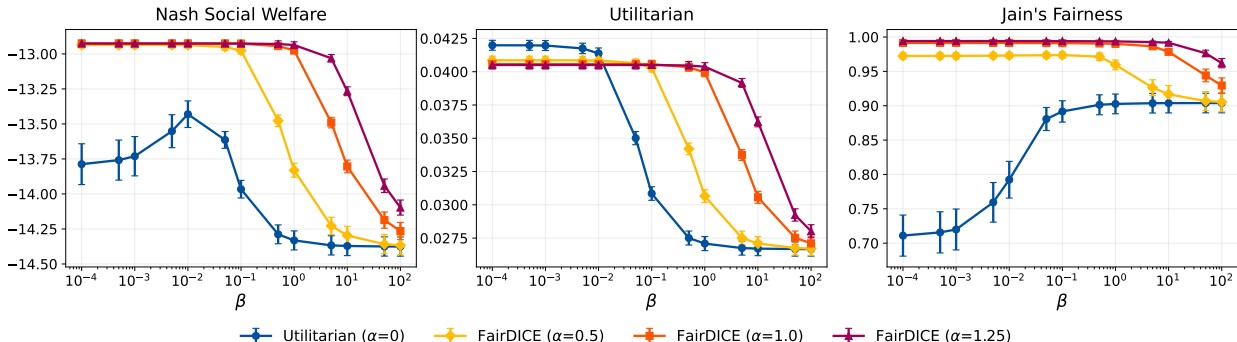

Figure 13: Our replication: Random MO-MDP metrics over a sweep of $\alpha$ and $\beta$ values.

## G Additional baseline comparisons for continuous environments

We evaluated FairDICE across a range of hyperparameters using 10 random seeds per configuration. For the reproduced model ('Rerun') experiments, we varied the $\beta$ values, while for the 'Fixed' variants, we swept over both $\beta$ and the gradient penalty coefficient. To ensure robust evaluation, we employed a split-seed protocol: the first 5 seeds were used exclusively for hyperparameter selection, identifying the configuration that maximized the mean Nash Social Welfare (NSW). The final reported results, including mean NSW scores and raw returns, were calculated using the remaining 5 held-out seeds for the selected configuration.

The resulting graphs, which combine both the baseline methods (as described in Appendix E) and the FairDICE results, are shown in Fig. 14, 15 and 16.

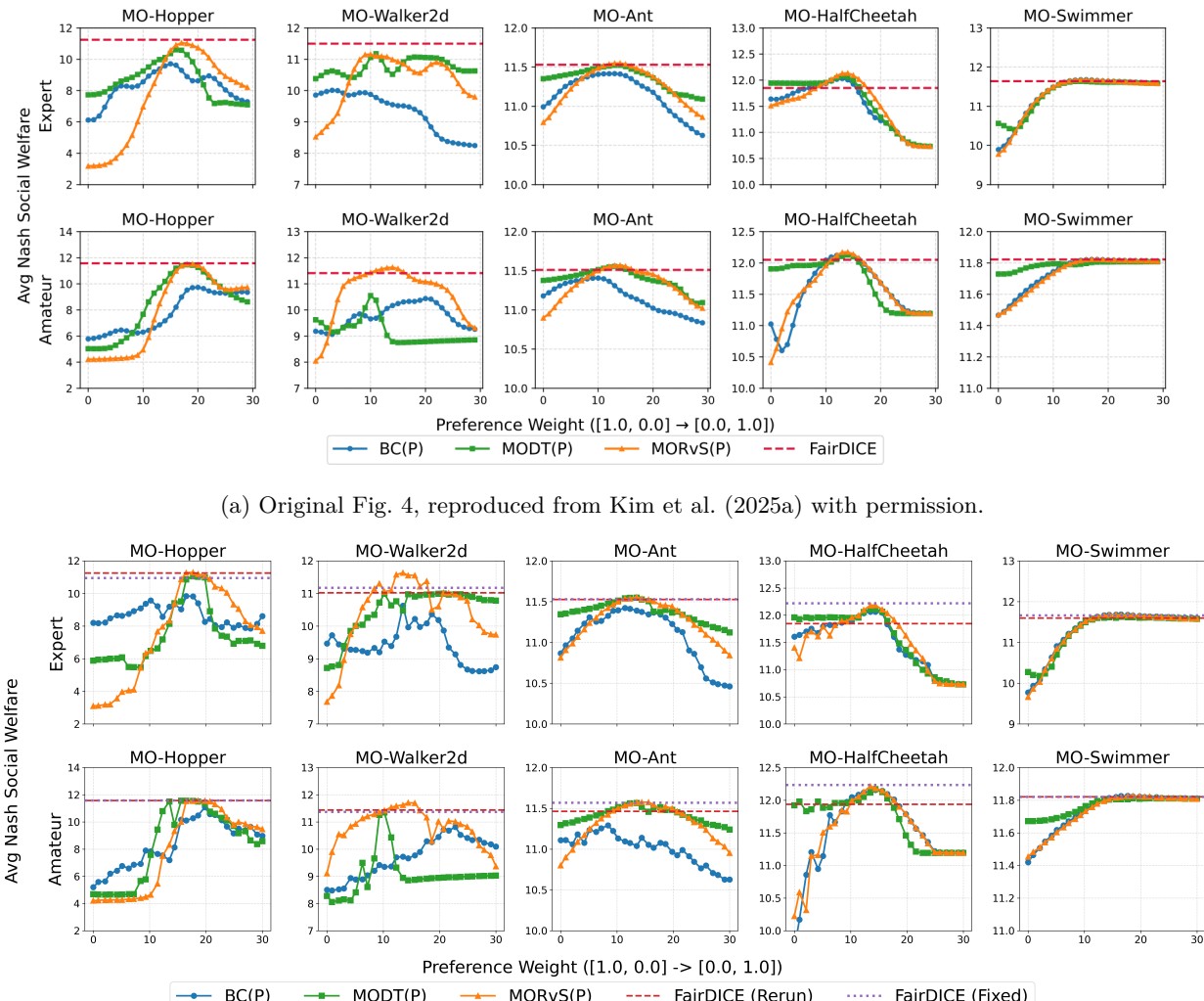

(a) Original Fig. 4, reproduced from Kim et al. (2025a) with permission.

(b) Reproduction

Figure 14: Nash social welfare across 30 linearly spaced preference weights on five two-objective tasks.

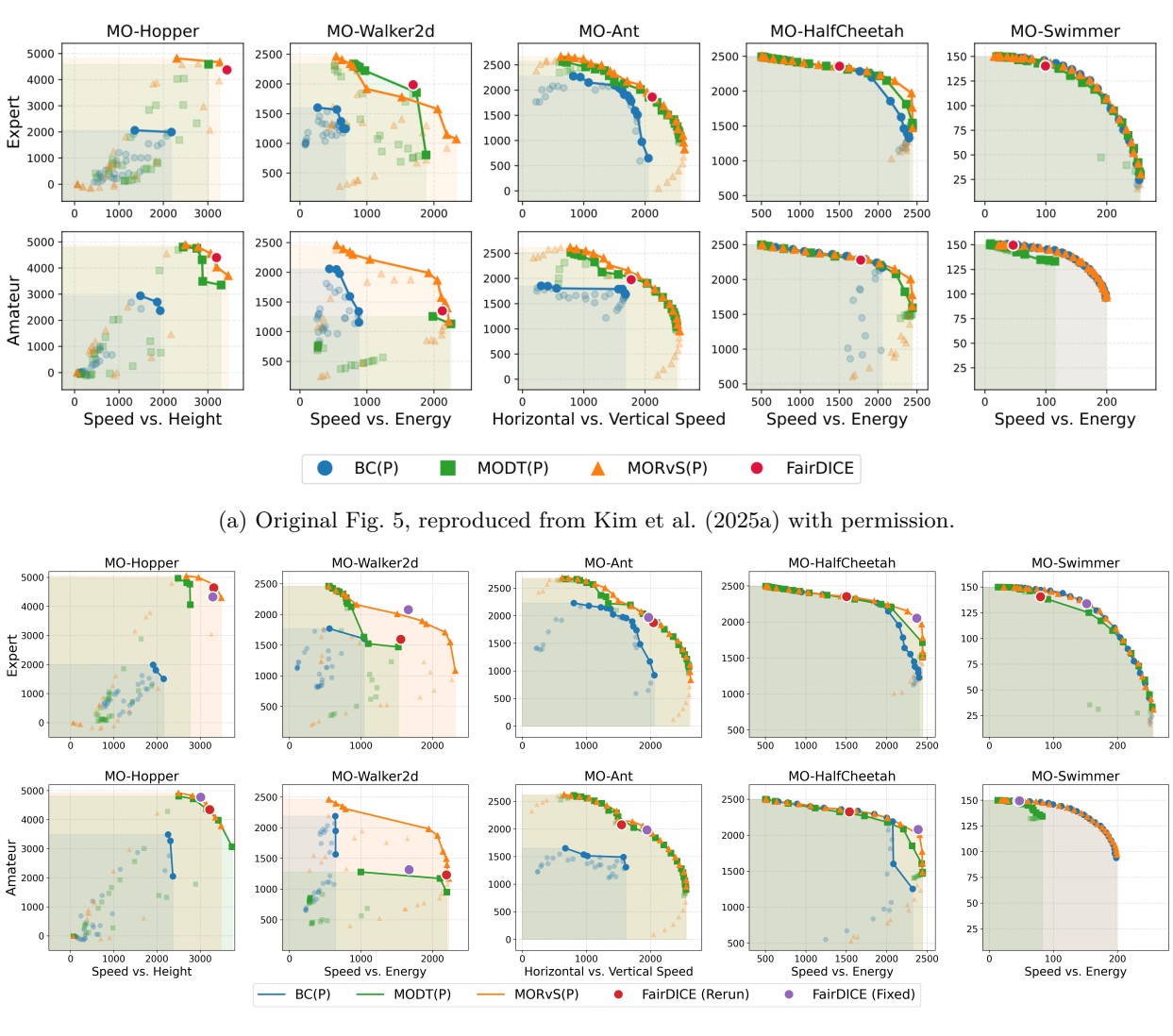

(a) Original Fig. 5, reproduced from Kim et al. (2025a) with permission.

(b) Reproduction

Figure 15: Raw return trade-offs on five two-objective tasks, with Pareto frontiers and dominated regions.

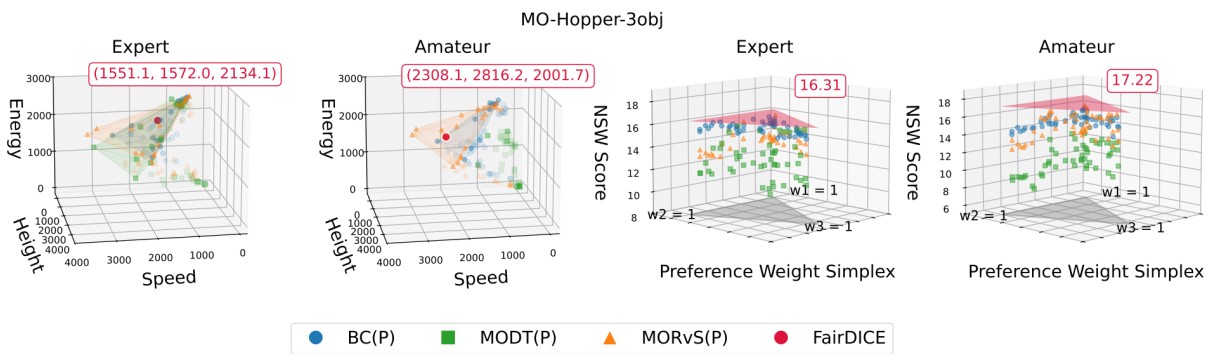

(a) Original Fig. 6, reproduced from Kim et al. (2025a) with permission.

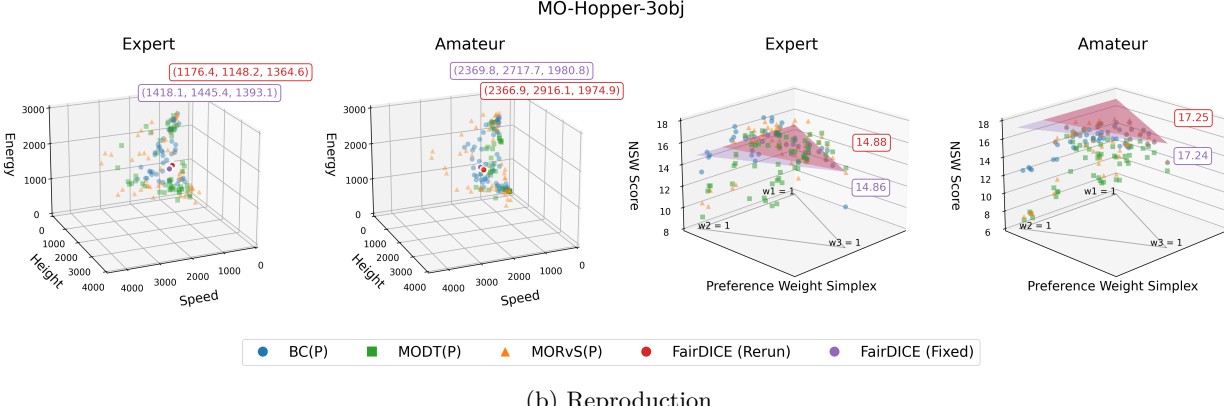

(b) Reproduction

Figure 16: MO-Hopper-3obj: raw returns and Nash social welfare (NSW) evaluations with three objectives.

# H    Effects of hyperparameter beta

Sec. 4.2.1 Fig. 3 provides only a selection of results for D4MORL tasks due to space limitations, but all results are available in Fig. 17 below. Similarly to Fig. 3 it is challenging to identify universal trends across all datasets, though typically a $\beta$ of 0.1 seems to work best.

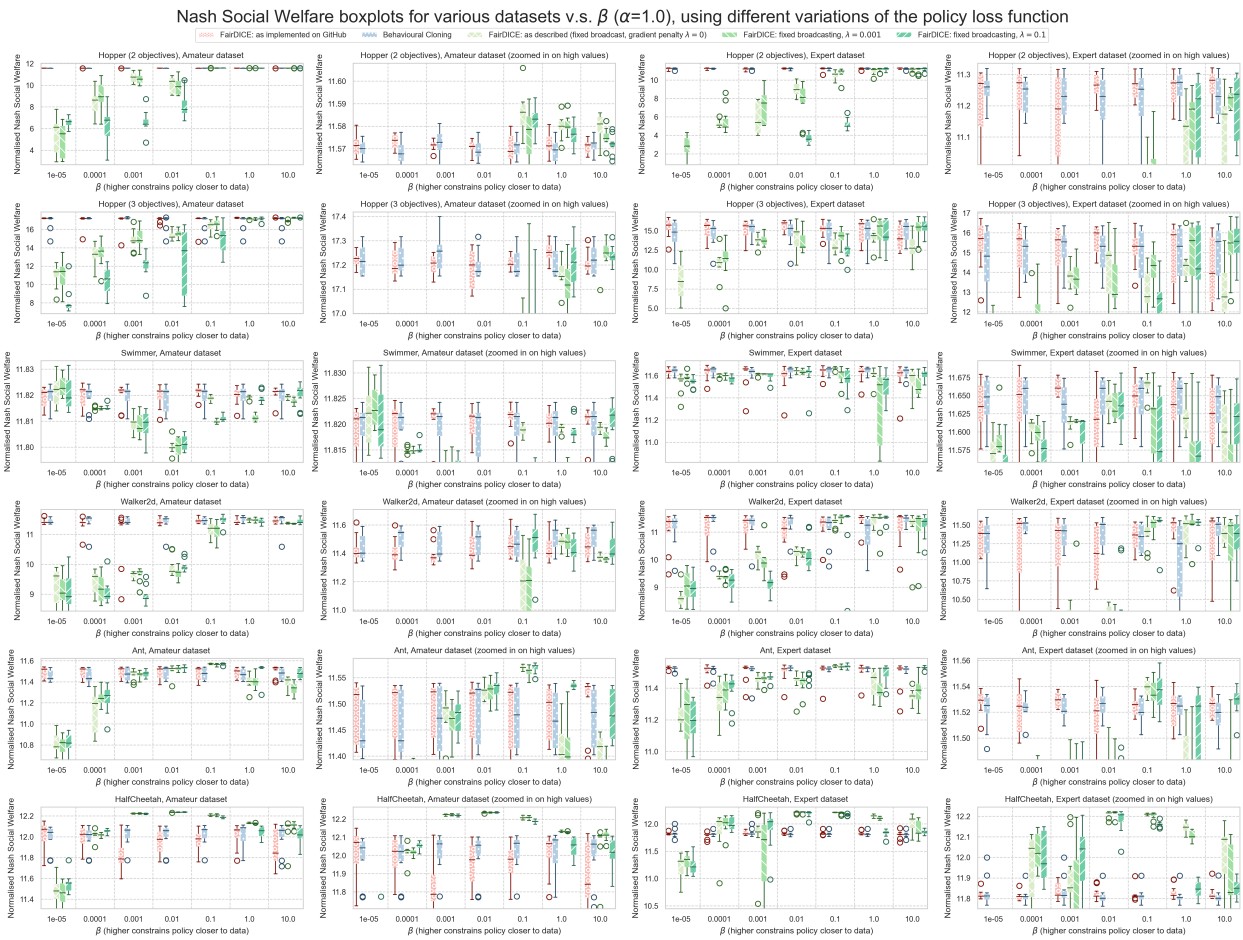

Figure 17: Complete version of Fig. 3, 10 seeds with 100 evaluations per seed.

## I Experimental hardware and Environmental impact

Experiments were run on four machines, which are described below in Table 3. Active hours were recorded and calculated using experimental logs, while wattage was measured as the average draw from the power outlet for local machines and estimated based on public information for clusters. Environmental impacts were estimated per location based on recent averages of grams of CO2eq per used kWh (Nowtricity, 2026).

| Used for Section(s) | GPU | CPU | Hours @ Wattage | g CO2eq/kWh | $\sim$ CO2eq |
|---|---|---|---|---|---|
| 4.2.2 | A100 ($\frac{1}{2}$ MIG) | Xeon Platinum 8360Y | 303h @ 250W | 385 | 29.2 kg |
| 4.2.1, 4.3.1, 4.3.2, 4.3.3, | RTX 5090 | i9-13900K | 273h @ 400W | 414 | 45.2 kg |
| 4.1.1, 4.1.2, 4.3.4 | A100 | AMD EPYC 7543 | 180h @ 300W | 385 | 20.8 kg |
| 4.3.2 | RTX 4060 | Ryzen 5-7600X | 3.5h @ 150W | 501 | 0.26 kg |

Table 3: Hardware specifications for running experiments and corresponding CO2eq estimates.

During training, Kim et al. (2025a) evaluate models every 100 steps, for a total of 100 intermediate evaluations (Kim et al., 2025c). This causes significant slowdown during training, as the intermediate evaluations require running MuJoCo simulations on the CPU, such that results in Kim et al. (2025a) require 20 minutes to train one policy. We instead opt to perform only 4 intermediate evaluations, and thereby accelerate training to 2-3 minutes per policy (depending on the environment, as some are more expensive to simulate). While Kim et al. (2025a) do not describe the total computational or environmental cost of their experiments, we can estimate that our experiments are 85-90% more computationally efficient, which would translate to a similar reduction in expected emissions (assuming similar g CO2eq/kWh as per LowCarbonPower, 2026).

## J BC-style FairDICE evaluated on increasingly unbalanced data

Kim et al. (2025b) demonstrated that FairDICE as implemented is robust to a degree of unbalanced trade-offs. This section verifies this and extends upon it by removing all trajectories where preference weights fall between 0.3 and 0.7, and 0.2 and 0.8, simulating an environment where the data is heavily biased towards a certain reward. These experiments are however conducted using FairDICE as implemented, meaning it behaves as behavioural cloning, not as FairDICE is intended to function. The results discussed here a thus **not reflective of FairDICE as described**, but serve mainly to **replicate more of the results displayed**[13].

Table 4 shows that the model largely retains a robust performance even when a substantial fraction of trajectories is removed. Across most environments and dataset qualities, filtering the trajectories results in a limited degradation in performance, despite removal of almost all data in some cases. Notably, the distribution of preference weights is not uniform across environments. Hopper-v2 and Ant contain only few imbalanced preference weights, meaning the more challenging datasets cannot be meaningfully evaluated, making the claim harder to assess.

| Environment | Dataset quality | Full | Traj. 0.4-0.6 filtered | | Traj. 0.3-0.7 filtered | | Traj. 0.2-0.8 filtered | |
|---|---|---|---|---|---|---|---|---|
| | | NSW | Traj. cut | NSW | Traj. cut | NSW | Traj. cut | NSW |
| Swimmer | Expert | $11.668 \pm 0.016$ | 24% | $11.581 \pm 0.080$ | 48.3% | $11.393 \pm 0.180$ | 72.5% | $10.752 \pm 0.441$ |
| | Amateur | $11.820 \pm 0.004$ | 24% | $11.811 \pm 0.003$ | 48.5% | $11.811 \pm 0.001$ | 72.7% | $11.817 \pm 0.001$ |
| Walker2d | Expert | $11.202 \pm 0.432$ | 34.9% | $9.845 \pm 1.227$ | 69.6% | $10.100 \pm 0.671$ | 94.3% | $10.967 \pm 0.005$ |
| | Amateur | $11.185 \pm 0.374$ | 35% | $11.342 \pm 0.010$ | 69.6% | $10.847 \pm 0.090$ | 94.1% | $11.114 \pm 0.012$ |
| Ant | Expert | $11.529 \pm 0.028$ | 43.1% | $11.298 \pm 0.041$ | 86.5% | $11.349 \pm 0.011$ | 100% | - |
| | Amateur | $11.484 \pm 0.048$ | 43.2% | $11.354 \pm 0.046$ | 86.5% | $11.454 \pm 0.002$ | 100% | - |
| HalfCheetah | Expert | $11.828 \pm 0.037$ | 43.6% | $11.709 \pm 0.032$ | 70.4% | $11.529 \pm 0.007$ | 92.6% | $11.345 \pm 0.012$ |
| | Amateur | $11.824 \pm 0.155$ | 43.9% | $11.650 \pm 0.029$ | 71.1% | $11.625 \pm 0.054$ | 92.9% | $11.386 \pm 0.009$ |
| Hopper | Expert | $11.170 \pm 0.176$ | 67.3% | $11.218 \pm 0.034$ | 100% | - | 100% | - |
| | Amateur | $11.570 \pm 0.003$ | 67.7% | $11.602 \pm 0.006$ | 100% | - | 100% | - |

Table 4: NSW comparison before and after filtering out balanced trajectories across all uniform environments (with two goals) and dataset qualities of FairDICE as implemented. All NSW results show the average NSW across 5 seeds. Traj. cut refers to the percentage of trajectories removed during filtering, '-' indicates NSW cannot be evaluated as no trajectories remain.

## K Fixed FairDICE using different D4MORL preference distributions

Beyond the coverage of preference weights, the shape of the preference-weight distribution may also influence performance. To isolate this effect, we evaluate FairDICE (as described) on two environments with both a 'narrow' and 'wide' distribution over preference weights[14], as shown in Table 5. It can be observed that the average NSW for all distributions and environments, while differing, is still very similar to performance on the uniformly distributed dataset, meaning the data distribution has limited impact on fair performance (although these distributions are themselves not very different, as can be seen in Appendix B.3.1).

| Environment | NSW (Uniform distribution) | NSW (Narrow distribution) | NSW (Wide distribution) |
|---|---|---|---|
| MO-Hopper | $10.820 \pm 0.367$ | $10.844 \pm 0.363$ | $11.074 \pm 0.143$ |
| MO-Swimmer | $11.587 \pm 0.070$ | $11.682 \pm 0.020$ | $11.618 \pm 0.039$ |

Table 5: FairDICE as described, with a $\beta$ of 1 and $\lambda$ of 0, trained on both the expert uniform, narrow and wide versions of Hopper-v2 and Swimmer, where the NSW is the average NSW across 5 seeds. The descriptions 'narrow' and 'wide' are from the D4MORL dataset from Zhu et al. (2023).

---

[13]This data was collected while we were uncertain about the nature of the broadcasting error in the provided code, and we provide these results for completeness to support the claim that, beyond the bug and missing code, the numerical results in Kim et al. (2025a) can all be replicated.

[14]This terminology originates from the D4MORL dataset — we visualise the distributions in Appendix Fig. 10.

## L   Likely typo in FairDICE derivation

On page 5 of Kim et al. (2025a), Sec. 5.1 provides a derivation for the core FairDICE algorithm. One of the terms in this derivation is $\sum_i u_i(k_i)$, which occurs positively at the top of the page and in the final loss, but negatively in an intermediate step: see Fig. 18. This appears to be a mistake in grouping terms.

Following the DICE-RL framework, we derive a sample-based optimization method from the Lagrangian dual of (P2-reg). We also highlight the challenges of extending (P1) directly to sample-based optimization and show how (P2) circumvents the issues. The Lagrangian is expressed as:

$$\max_{d \geq 0, k} \min_{\nu, \mu} L(\nu, \mu, d, k) := \sum_i \mu_i \left( \sum_{s,a} d(s,a) r_i(s,a) - k_i \right) - \beta \sum_{s,a} d_D(s,a) f \left( \frac{d(s,a)}{d_D(s,a)} \right)$$

$$\boxed{+} \quad \boxed{+ \sum_i u_i(k_i)} + \sum_s \nu(s) \mathcal{F}_d(s)$$

We reparameterize the stationary distribution as $d(s,a) = w(s,a) d_D(s,a)$ and express the dual in terms of the importance weights $w$, using the identity $\sum_s \nu(s) \sum_{\bar{s},\bar{a}} T(s|\bar{s},\bar{a}) d(\bar{s},\bar{a}) = \sum_{s,a} d(s,a) \sum_{s'} T(s'|s,a) \nu(s')$. This yields the following optimization problem:

$$\max_{w \geq 0, k} \min_{\nu, \mu} \mathbb{E}_{s \sim p_0}[(1-\gamma)\nu(s)] + \mathbb{E}_{d_D}[w(s,a) e_{\nu,\mu}(s,a) - \beta f(w(s,a))] \boxed{- \sum_i (\mu_i k_i + u_i(k_i))}$$

where $e_{\nu,\mu}(s,a) = \sum_i \mu_i r_i(s,a) + \gamma \sum_{s'} T(s'|s,a) \nu(s') - \nu(s) \; \forall s,a$.   $\boxed{-}$

Applying the Lagrangian dual directly to the regularized (P1) retains expected returns inside the concave functions $u_i(\cdot)$, preventing direct use of importance sampling. Moreover, a naive estimator such as $\sum_{s,a} d_D(s,a) \sum_i u_i(w(s,a) r(s,a))$ introduces bias, violating the validity of importance-weighted estimation.

We further simplify the optimization by reducing parameters. Using strong duality of (P2-reg), we switch the optimization order to $\min_{\nu,\mu} \max_{w,k}$ and derive closed-form solutions for $w$ and $k_i$ from first-order conditions:

$$w_{\nu,\mu}^*(s,a) = \max \left( 0, (f')^{-1} \left( \frac{e_{\nu,\mu}(s,a)}{\beta} \right) \right) \; \forall s,a, \quad k_{i,\mu}^* = (u_i')^{-1}(\mu_i) \; \forall i$$

Substituting the closed-form solutions into the Lagrangian dual yields the final optimization, which defines the loss function of our offline algorithm, FairDICE (Fair MORL via Stationary Distribution Correction):

$$\min_{\nu,\mu} \mathbb{E}_{s \sim p_0}[(1-\gamma)\nu(s)] + \mathbb{E}_{(s,a) \sim d_D} \left[ \beta f_0^* \left( \frac{e_{\nu,\mu}(s,a)}{\beta} \right) \right] \boxed{+ \sum_i u_i^*(-\mu_i)} \quad \boxed{+} \quad (4)$$

where $f^*(y) := \max_{x \geq 0} xy - f(x)$ and $\boxed{u_i^*(y) := \max_x xy + u_i(x)}$ are convex conjugate functions. Solving (4) gives the optimal stationary distribution $d^*(s,a) = w_{\nu^*,\mu^*}^*(s,a) d_D(s,a)$. In finite domains, the optimal policy is directly recovered via $\pi^*(a|s) = d^*(s,a) / \sum_a d^*(s,a) \; \forall s,a$.

Figure 18: Unexpected change of sign in the FairDICE derivation, page 5 of Kim et al. (2025a)

From a theoretical point of view, the $u_i(k_i)$ term occurring negatively would not make sense, as this would allow for a trivial solution as $\mu$ approached **0** where all rewards are ignored. From a practical point of view, a handful of small-scale experiments verify that implementing the loss with $u_i$ negated will indeed result in the optimiser selecting a trivial solution for $\mu$, which results in all terms of $w^*$ becoming near-identical and the algorithm reducing to behaviour cloning (even with the broadcasting error fixed in code).

