# OpenReview forum: "[Re] FairDICE: A Fair Tradeoff in Multi-objective Offline RL"
_TMLR — Accepted by TMLR_

### Review · Reviewer_na2A · 2026-03-24

**Summary Of Contributions:**

This submission is a reproducibility study of "FairDICE: Fairness-driven offline multiobjective reinforcement learning." The authors identify two key flaws in the original empirical results: 1) various hyperparameters and implementation choices were not disclosed, and 2) a bug in programmatic implementation of the algorithm made it equivalent to behavior cloning. The latter led to an incorrect conclusion in the original publication that the algorithm was insensitive to a particular hyperparameter, which has no impact on the behavior-cloning-equivalent implementation. The submission then extends the original paper by studying four settings: 1) negative returns, 2) biased datasets, 3) high-dimensional rewards, and 4) image-based observations.

######## Strengths ########

- The work correctly identifies a flaw in the original paper. The authors detail their continued communications with the original authors, which lends credence to their finding of the implementation error.
- The manuscript provides precise details of how the experiments are conducted and the steps taken to ensure fairness of comparison with the original implementation and with baselines.
- The work provides experiments to empirically validate claims made by the original authors in their rebuttals. These types of claims often go untested, so it is commendable that the authors took steps to check their validity.

######## Weaknesses ########

- The manuscript is a lot closer to a technical report than a scientific publication. It provides careful implementation details and results. The paper would be a lot stronger if it provided new scientific insights. Some examples where such insight seems to be missing:
    - Sec 4.1
        - There is no analysis of why "minor differences remain" between the authors' and the original paper's results in Random MOMDP. Is this due to a discrepancy in implementation, hyperparameter selection?
    - Sec 4.2
        - It is not clear why the fixed and original versions of FairDICE both yield performance on the Pareto frontier.
    - Sec 4.3
        - The explanation in Sec 4.3.1 for why FairDICE performs well even with unnormalized rewards is insufficient. Why does it only work if the expected returns are positive? The $k_i$'s in Footnote 9 are undefined and so the clarification is not immediately helpful to readers.
        - "due to a lack of time we are unable to reproduce this experiment using the corrected loss" — does this imply that the loss from Appendix H in the original paper have the same implementation flaw due to the incorrect broadcasting? If not, then what is the necessary correction? What even is the loss in Appendix H and how does it relate to the loss used in the rest of the work? What is the use of this experiment if using an incorrect loss? Also, a lack of time is not a great excuse for submitting incorrect/incomplete results, especially for a model like TMLR without a deadline.
        - Why might FairDICE be less sensitive to hyperparameter choices in the experiments in Sec 4.3.4? This is mentioned but not analyzed
- There is little motivation provided for why it is meaningful to extend FairDICE once the authors demonstrate that the original implementation is flawed and that even the fixed implementation is not meaningfully better than BC in continuous action spaces.
    - Given the results in Fig. 3, the authors seem to debunk the empirical usefulness of FairDICE for (these) continuous domains, since simpler BC outperforms it in most cases with no hyperparameter tuning. With that, could the authors expand on why there is value in expanding evaluations past this point? I would understand their use if those evaluations were geared toward understanding why discrepancies arise between any theoretical claims made in FairDICE and the results obtained with the (corrected) implementation. But simply showing more performance measures fails to provide insight into the failures of Fig. 3—it becomes more a plain reproducibility study.
    - Moving on to "Extensions to the original paper" without a proper understanding of the reasons why the original and fixed implementations perform comparably or how the original implementation achieves high NSW appears rash.

**Additional Comments:**

The following points are provided as feedback to hopefully help better shape the submitted manuscript, but will not impact my recommendation in a major way.

Title
- The title is not informative of what the paper is about. Mentioning "multiobjective offline RL" (at a minimum) and "reproducibility study" (ideally) would make it a lot more informative

Intro
- There's work on offline proxies for online performance, which could be used for hyperparameter optimization. The authors may wish to soften the claim that hyperparameter tuning is "necessarily online"
- "experiments in discrete environments" — what about continuous?

Sec 2.3
- "while reaching either objective more often..." — is this restricted to two objectives (as implied by the use of 'either') or broadly applies to a collection of objectives?
    - Extension 3.2 also hints at only two objectives given the comment about "preference weights far from $[0.5, 0.5]$"
    - Sec 3.1 confirms that for the original continuous experiments there are indeed two objectives. The authors should explicitly clarify this.
- "changing $\beta$ interpolates between..." — $\beta$ was not introduced in Sec 2.2

Sec 4.1
- What is Jain's fairness?
- "higher Jain's fairness at the cost of slightly lower utilitarian welfare" — is the use of 'slightly' due to the y-axis scale? How does one interpret scale in utilitarian vs Jain's fairness?


Typos/style/grammar
- Sec 3.4 -- Nash Social Welfare should not be spelled out since its acronym has been introduced (and used extensively)
- Sec 4.1.2 -- "Some minor differences remain[.] Utilitarian..."
- Fig. 3 -- Why is the Ant figure so much larger than other domains?

**Audience:**

No

**Audience Explanation:**

This technical report focuses primarily on identifying claims within the original FairDICE paper that hold or fail to hold empirically. While these are informative and useful to the community, they mostly suggest that the original paper should either be revised or retracted. While a reproducibility study could certainly be of interest to at least individual members of the TMLR community, the current manuscript should dive deeper into the analyses throughout Sec. 4, to identify not just the results but the reasons behind them. These additional insights would potentially be of interest.

**Claims And Evidence:**

No

**Claims Explanation:**

The title, Abstract, and Introduction all suggest that the reproducibility study in the submission will explore whether a gap exists between theory and practice (e.g., "We find that many theoretical claims hold..."). However, the authors later state that the submission "[does] not focus on examining the mathematical validity...". The manuscript later refers to theoretical results (e.g., "consistent with theoretical predictions" in Sec 4.1.1), but these theoretical results are not stated, explained, or analyzed in light of the empirical findings.

Other claims, in particular the identification of flaws in the original paper and the extensions are well supported.

**Requested Changes:**

Key changes:
1. Adjusting the claims about the gap between theory and practice (see box under "Are the claims made in the submission supported by accurate, convincing and clear evidence?")
2. Providing additional insights about the results (see box under "Would at least some individuals in TMLR's audience be interested in knowing the findings of this paper?" and "Summary Of Contributions -> Weaknesses").

---

> ### Author Response · Authors · 2026-03-30
> **Reply 1/3: Regarding minor changes and typos**
>
> Thank you for your very detailed review of our paper, we sincerely appreciate your feedback.
> We would like to discuss each point of feedback provided, but the most important parts of our response are 2B, 2F, 3A and 3B.
>
> Firstly, we would like to discuss some of the minor changes we have made to our submission in light of your “Additional Comments”:
>
> **1A (Title)**: We realise that our title may indeed be unclear, especially when presented in a context without keywords. We have therefore chosen to instead use the title “[Re] FairDICE: A Fair Tradeoff in Multi-objective Offline RL”. Regarding “Reproducibility study”, we were under the impression that the “[Re]” title tag was often used in direct reproducibility studies to distinguish them from their counterparts.
>
> **1B (Intro.hyperparameters)** We have adjusted our claim to state that it is challenging but not impossible to perform offline hyperparameter selection.
>
> **1C (Intro.continuous)** Our intention here was to emphasise that Kim et al. use discrete (toy) environments to verify theoretical claims, and that we also primarily rely on these discrete environments for this purpose. However, continuous environments also partially support the theoretical claims made, so we have rephrased the sentence to avoid the possible alternate interpretation.
>
> **1D (Background.either objective)** We have rephrased the claim to remove “either”.
>
> **1E (Background.beta)** Due to several revisions of our Background section, it appears we removed all explanation of hyperparameter beta from the final draft. We have now added back a sentence at the end of page 2, thank you for spotting this.
>
> **1F (Results.Jain)** We have added a brief explanation in our method section: it is the sum of rewards squared divided by the sum of squared rewards.
>
> **1G (Results.slightly)** We appreciate this observation. To clarify: utilitarian reward and Jain's fairness index are independent metrics, and we would ideally like both to be high. This naturally gives rise to a tradeoff curve as alpha varies, where an "acceptable" tradeoff might be identified at an elbow point. We acknowledge that our use of "slightly" was imprecise; our intended meaning was that the drop in utilitarian performance from increasing alpha is small relative to the drop caused by increasing beta, making the fairness gain a comparatively cheap tradeoff. We have revised the text to make this comparison explicit rather than relying on the subjective term.
>
> **1H (Method.NSW)** We have used the acronym NSW in more places after its definition.
>
> **1I (Method.punctuation)** , is now a ;
>
> **1J (Results.Fig3)** We originally believed a “spotlighted” subgraph useful to guide the reader in understanding the graph, but after some consideration we have decided that providing more information to the reader would likely be better in this case; the tiling has been readjusted to include four medium-sized graphs in Fig. 3.

---

> > ### Author Response · Authors · 2026-03-30
> > **Reply 2/3: Regarding major changes and questions**
> >
> > Secondly, we would like to address the main weaknesses mentioned in your review:
> >
> > **2A (4.1.MOMDP)** Qualitative results mostly hold, and differences in exact results could be due to a still slightly different setup than the one used originally (the authors haven’t shared MO-MDP code, only MO-FourRooms so we re-implemented MO-MDP based on it)
> >
> > **2B (4.2.Pareto)** It is indeed noteworthy that the original FairDICE (which is BC) produces policies on the Pareto frontier, and we have therefore added some additional text to explain this in Sec. 4.2.2. Repeating it here, it is because 1a) we suspect most intermediate-weighted policies trained on D4MORL behave similarly (e.g. they must all learn to move forward in a balanced manner), 1b)  ‘amateur’ versions of D4MORL are not generated by amateur policies, but merely noised versions of expert policies which BC can learn to de-noise, and 2) Kim et al. use training tricks (cosine LR schedule and orthogonal initialisation) which improve performance. In very small-scale experiments, we confirm that disabling either of the training tricks or using a slightly different evaluation technique (sampling actions instead of taking the mean of the action distribution predicted by the policy) significantly degrades performance, yielding results no longer on the Pareto front. However, this primarily shows that D4MORL may be a suboptimal benchmark for evaluating techniques such as FairDICE, and emphasizes the importance of alternative evaluation environments.
> >
> > **2C (4.3.Unnormalised)** We have expanded the footnote slightly, to provide the context we believe is relevant. If you believe the reader would be well-served by further explanation beyond this addition, we can also add a dedicated section in the appendix.
> >
> > **2D (4.3.Lack of Time)** We fully understand that this is not entirely appropriate, but due to practical considerations it is challenging for us to perform large-scale experiments now that the initial period of experimentation has elapsed. We agree that this section may not be the most relevant, and we have therefore moved it to the appendix in favour of additional text for other sections.
> >
> > **2E (4.3.Hyperparameters)** We hypothesise that Minecart's lower sensitivity to beta stems from two factors. First, the dataset was collected from PPO-trained policies, which may introduce a distributional bias that constrains the range of learned behaviours. Second, the reward dimensions in Minecart are coupled: fuel consumption correlates with ore collection, so policies optimising for different weightings tend to behave similarly, reducing the impact of varying beta. We note, however, that with only three seeds per configuration, the observed low sensitivity could also partly reflect limited statistical power.
> >
> > **2F (Why expand evaluations?)** As mentioned above, we believe that the overly high performance of BC on D4MORL suggests that D4MORL may not be the most representative benchmark for this purpose, especially since earlier results in discrete environments did show clear improvements over what BC would be able to achieve. As such, our motivation for the two additional environments in Sec. 4.3.3 and Sec. 4.3.4 is to better understand when FairDICE can improve over the data; we seek to accomplish this by creating datasets where there is no opportunity for BC to deduce an underlying expert policy (as was unfortunately possible with the amateur dataset being composed of noised expert actions). We then show that FairDICE can still improve over the data policy in these cases, when BC would not be able to. To clarify this motivation, we added a brief introductory paragraph to Sec. 4.3.
> >
> > **2G (Moving on)** We hope that our current draft does provide the reader with a proper understanding of why the original and fixed implementation perform comparably.

---

> > ### Author Response · Authors · 2026-03-30
> > **Reply 3/3: Regarding support of claims made and possible audience**
> >
> > **3A: Support of claims made**
> > Our replication work indeed primarily focuses on the empirical and practical parts of FairDICE, with theoretical statements (primarily Claim 1.1 and Claim 1.2) being supported by experimental evidence but not explained in detail or analysed. This reflects our focus on empirical replication of the paper, but we realise that it could have been phrased more clearly. As such, we have updated our abstract and conclusion such that we “find that many theoretical claims are supported”. We hope that this formulation better reflects our intent, and would be more clearly supported by our work.
> >
> > **3B: Audience**
> > Regarding a possible audience: when writing the original paper and especially these revisions, we have endeavoured to provide information relevant to three groups (in order of increasing scope)
> > Those interested in MORL and fairness in RL might be interested in knowing more about how FairDICE performs.
> > Those developing multi-objective RL algorithms might further be interested in our findings on the possible limitations of D4MORL as an offline RL benchmark (in light of the Pareto-efficient performance of BC).
> > General RL readers might be interested in reading about challenges encountered during reproduction, to improve the reproducibility of future work
> >
> > **3C: Final notes**
> > We would again like to thank you for your time, and we would appreciate any feedback regarding our revised submission or a reconsideration of the initial judgements regarding the support of claims made and a possible audience.

---

> ### Author Response · Authors · 2026-04-11
> **Discussion period lasts until tomorrow**
>
> We would again like to thank you for your helpful feedback and comments on our submission, much of which we have integrated into our second revision. If you have any comments or feedback on this revision, or on our answers to your questions, we would greatly appreciate it.
>
> Should you have any such comments, we would of course wish to remind you that the official discussion period ends tomorrow (Apr 12).
> If you do not have any comments, we hope that our revisions or explanations have sufficiently addressed all weaknesses mentioned.

---

### Review · Reviewer_eosH · 2026-03-28

**Summary Of Contributions:**

The authors perform a replication study in which they assess claims made regarding the FairDICE algorithm. They assess theoretical claims related to the algorithm, and find that they hold in general. However, they find a bug in the code used to generate the results in the original FairDICE paper, and show how this bug invalidates some of the empirical claims made regarding FairDICE. After rectifying the bug, the authors show that FairDICE is still empirically viable and can scale to more complex environments.

**Audience:**

Yes

**Audience Explanation:**

This paper would be of interest for individuals interested in offline and multi-objective RL.

**Claims And Evidence:**

Yes

**Claims Explanation:**

The authors make three claims in this paper:

1) That the theoretical properties of FairDICE hold in general.
2) That the empirical claims made in the original FairDICE paper are invalid.
3) That FairDICE is still empirically viable and can scale to more complex environments.

I find that all three claims are adequately supported in the paper. In terms of the first claim, the results shown in Section 4.1 show, empirically, that the claimed theoretical properties of FairDICE hold. In terms of the second claim, the rigorous results and discussion included in Section 4.2 adequately supports the claim that the original empirical results related to FairDICE do not hold since there was a bug in the original authors’ code. Finally in terms of the third claim, I find the results in Section 4.3 to be convincing and adequate.

**Requested Changes:**

From a writing perspective, the paper is very well-written and I have no concerns in this regard. The figures included in the text are also clear and easy to understand.

All in all, I have no concerns regarding this paper and I believe it to be in a state worthy of publication. The authors were very deliberate and precise in their arguments. Moreover, I find that the authors were fair and acted in good faith when it came to representing the claims made by the authors of the original FairDICE algorithm. All things considered, I find this paper to be a well-executed and high-quality replication study.

My one question to the authors is this: Do you see a need to perform a more rigorous analytical (i.e., theoretical) review of the work done by the original authors of FairDICE beyond what is included in Appendix A of this paper?

---

> ### Author Response · Authors · 2026-03-30
> **Reply to review by Reviewer eosH**
>
> Thank you for your review of and positive comments on our reproducibility study.
>
> Regarding your question, it is indeed the case that we have been hesitant to include more details or analysis of the mathematics underlying FairDICE and the DICE framework itself. This is primarily to enable our reproducibility study to focus on the empirical and practical aspects of FairDICE. We had previously performed a detailed review of the derivation and found it self-consistent (excluding a typo, which we describe in Appendix L) and in line with related work. We have added a sentence stating exactly this in our Method, to clarify our position.
>
> However, we understand the need for a clear explanation and motivation of a method when discussing it, hence the intuitive explanations in the Background section. We have now also added recommended reading to the end of Appendix A, directing the reader to papers we found informative around DICE.

---

### Review · Reviewer_NkGw · 2026-03-29

**Summary Of Contributions:**

This paper is a replication and experimental study, not a paper that proposes a new method for offline multi-objective reinforcement learning (MORL). Its goal is to evaluate whether the claims made by the original FairDICE paper are correct in practice.

In offline MORL, an agent must learn only from a fixed offline dataset while balancing multiple potentially conflicting objectives. A key challenge is to achieve a fair trade-off among objectives, rather than over-optimizing one objective at the expense of others.
FairDICE is the algorithm being evaluated. It extends OptiDICE to offline MORL by learning the weights over objectives automatically, instead of requiring them to be chosen by hand. The purpose is to optimize fairness-oriented objectives such as Nash Social Welfare.

The replication paper finds that the public implementation of FairDICE contains an implementation bug in the policy loss for continuous environments. Because of an incorrect broadcasting operation, the learned importance weights are ignored, and the method effectively reduces to standard behavior cloning in those settings.

The main contribution of this paper is to show a gap between FairDICE theory and practice. Its main findings are:
1. Discrete setting:
The main theoretical claims of FairDICE are largely supported in the discrete experiments. The reproduction shows that FairDICE can learn more balanced policies, and the effects of the hyperparameters are broadly consistent with the original theory. In particular, changing $\beta$ behaves as expected: a small $\beta$ allows the learned policy to deviate more from the behavior policy, while a large $\beta$ pushes the policy back toward behavior cloning.
2. Continuous setting:
The original continuous-environment results are not reliable because they were produced by the buggy implementation. After correcting the bug, FairDICE no longer appears robust across values of $\beta$; instead, its performance becomes highly sensitive to hyperparameter choice. Although corrected FairDICE can still perform well in some cases, its results are unstable across datasets, which weakens the original claim that it consistently outperforms or matches existing offline MORL baselines.
3. Extended Exploration:
The author gives four extended exploration environment setting of FairDICE. The author runs FairDICE in different environment settings and observes the performance.

Overall, the paper concludes that FairDICE is a theoretically interesting method, but its practical effectiveness is much more limited than the original paper suggests, especially in continuous environments.

**Audience:**

Yes

**Audience Explanation:**

This paper provides comprehensive experiments on the performance and hyperparameter sensitivity of FairDICE, which can help practitioners understand its strengths and weaknesses.

**Broader Impact Concerns:**

There is no broader impact concerns as all experiments not involve human objects.

**Claims And Evidence:**

Yes

**Claims Explanation:**

This paper conducts many experiments to examine the claims made by FairDICE. Each claim comes with an experiment to support it.

**Requested Changes:**

There are several points that can help to improve the impact of this paper:
1. The original FairDICE comes with a theoretical analysis. It would be better that the authors leave some comments about the soundness of FairDICE.
2. The paper shows that FairDICE is highly sensitive to $\beta$ in continuous settings. It would be very helpful if the authors could propose practical heuristics for choosing $\beta$, or analyze patterns across environments.
3. The paper uses Nash Social Welfare (NSW) as the main metric. It would be useful to discuss limitations of NSW, and include or comment on alternative fairness metrics (e.g., min-max, Jain’s index interpretation).
4. The paper notes that tuning \beta may require online validation, which contradicts the offline RL setting. It would be useful to explicitly discuss whether FairDICE is still suitable for strictly offline scenarios, or if it should be considered semi-offline (with validation).

---

> ### Author Response · Authors · 2026-03-30
> **Reply to review by Reviewer NkGw**
>
> Thank you for your review and recommendations for improvement. We would like to briefly go through each of the four recommendations:
>
> **A (soundness)** We had previously reviewed the derivation of FairDICE and found no errors, save what is effectively a typo in an intermediate step (described in the Appendix L). We have now added an additional sentence to comment on our position regarding the validity of the FairDICE derivation.
>
> **B (choosing beta)** This is indeed the crux of the issue: in Appendix H we provide results for all values for beta in all environments, but there does not seem to be any universal pattern. For example, Hopper environments have a general downward trend in performance as beta decreases, but Swimmer environments have a U-shape where both high and low values for beta perform better than intermediate values, while HalfCheetah has a cap(∩)-shaped trend. As such, the most general advice we can give is that “a beta of 0.1 seems like a good guess, but verification/tuning is likely necessary.”
>
> **C (NSW alternatives)** The primary reason we focus on NSW performance, is that this is also the metric being optimised by FairDICE (by virtue of log being the nonlinearity applied). We also replicate experiments by the original authors on using intermediate fairness metrics (alpha-fairness) in discrete environments, and show that changing our target metric towards min-max fairness (by increasing alpha) also increases Jain’s fairness, but that the base configuration of alpha=1 (equivalent to log) also generalises to decent performance on Jain’s fairness. While it would indeed be interesting to extend this to a full comparison on how optimising different kinds of nonlinear fairness functions affects model performance on other fairness metrics (i.e. which fairness metrics, when used as target, might lead to the best ‘fairness generalisation’), we believe this is not entirely within the scope of our current replication paper, but perhaps interesting for future work.
>
> **D (semi-offline?)** We note in the Introduction and Discussion that this unreliable performance makes FairDICE hard to deploy in truly offline environments, but as Reviewer na2A noted it might be possible (and perhaps more so in the future) to use offline proxy metrics as an alternative to enable fully-offline tuning of beta. However, we would indeed consider it a “semi-offline” method as of right now.

---

### Author Response · Authors · 2026-03-30
**We appreciate the reviews, questions and feedback; some modifications made**

We would like to thank all the reviewers for their time in going through our paper, as well as their questions and remarks on how to improve our manuscript. We have placed specific replies to the questions or comments made in each review, and have made adjustments to the main text. To summarise these changes, we have:

- Changed the title to better reflect the subject matter.
- Added additional clarifications regarding hyperparameter beta and our position on the soundness of FairDICE (namely that it is correct, to the best of our understanding) to the Background and Method.
- Added additional analysis regarding the cause of the original FairDICE (BC) performing well on D4MORL (we would like to thank Reviewer na2A for attending us on the noteworthiness of this).
- Slightly edited some transitions in the Results and Discussion section to more clearly present our conclusions.
- Moved the first half of Sec. 4.3.2 to the Appendix, to make room for the aforementioned content.

We hope that our replies have clarified all concerns raised by the reviewers, and we remain open to further questions or feedback during this discussion period.

---

### Decision · Action_Editor_sXC6 · 2026-05-07

**Recommendation:** Accept with minor revision

**Additional Comments:**

It is required that authors incorporate the comments of the reviewers in the revision.

**Audience:**

Yes

**Audience Explanation:**

Both offline RL and multi-objective RL (MORL) are active research fields individually. Offline MORL may rather have  narrow audience, as noted by Reviewer na2A. From a practical viewpoint, however,  many real-world problems are multi-objective optimization with trade-off among objectives. In this sense, fairness is important and FairDICE seems a viable solution in this direction in future. It seems that authors set up an extensive reproducibility study on FairDICE and showed its potential.

**Claims And Evidence:**

Yes

**Claims Explanation:**

This is a reproducibility study of FairDICE, a paper published in NeurIPS 2025. The submitted paper claims that the original FairDICE paper's implementation has a bug in their code and the authors corrected this bug and show the applicability of FairDICE to more complex environments and high-dimensional rewards. The authors properly set their simulations to support their claims.